# Model simulations unveil the structure-function-dynamics relationship of the cerebellar cortical microcircuit

Robin De Schepper [1], Alice Geminiani[1], Stefano Masoli [1], Martina Francesca Rizza[1], Alberto Antonietti [1], Claudia Casellato [1,3✉] & Egidio D'Angelo [1,2,3✉]

The cerebellar network is renowned for its regular architecture that has inspired foundational computational theories. However, the relationship between circuit structure, function and dynamics remains elusive. To tackle the issue, we developed an advanced computational modeling framework that allows us to reconstruct and simulate the structure and function of the mouse cerebellar cortex using morphologically realistic multi-compartmental neuron models. The cerebellar connectome is generated through appropriate connection rules, unifying a collection of scattered experimental data into a coherent construct and providing a new model-based ground-truth about circuit organization. Naturalistic background and sensory-burst stimulation are used for functional validation against recordings in vivo, monitoring the impact of cellular mechanisms on signal propagation, inhibitory control, and long-term synaptic plasticity. Our simulations show how mossy fibers entrain the local neuronal microcircuit, boosting the formation of columns of activity travelling from the granular to the molecular layer providing a new resource for the investigation of local microcircuit computation and of the neural correlates of behavior.

[1] Department of Brain and Behavioral Sciences, University of Pavia, Via Forlanini 6, 27100 Pavia, Italy. [2] IRCCS Mondino Foundation, Brain Connectivity Center, Via Mondino 2, 27100 Pavia, Italy. [3] These authors jointly supervised this work: Claudia Casellato, Egidio D'Angelo. ✉email: claudia.casellato@unipv.it; egidiougo.dangelo@unipv.it

The relationship between structure, function and dynamics in brain circuits is still poorly understood posing a formidable challenge to neuroscience[1]. The core of the issue is how to deal with the distribution and causality of neural processing across multiple spatio-temporal scales. While experimental measurements remain essential, they can now be supported and complemented by realistic computational models. In principle, such models could take into account multi-modal datasets representing morphology, connectivity and activity of different cell populations and make it possible to simulate the propagation of microscopic phenomena into large-scale network dynamics[2–4]. These models can incorporate a broad range of biological data becoming highly constrained and providing the best proxies of the corresponding natural circuits. Eventually, once properly configured and validated, these models can generate their own ground-truth by binding the many parameters, provided by independent measurements and intrinsically prone to experimental error, into a coherent construct, and can be used to test various functional hypotheses[5] using specific simulations platforms, like NEURON[6] and NEST[7]. There are several examples of advanced computational models that have been mostly developed to simulate activities in the cerebral cortex[8–10]. Here we have developed a framework, the Brain Scaffold Builder (BSB), to cope with the organization of the cerebellar network.

The cerebellar cortical microcircuit has inspired foundational theories on brain functioning[11] but still challenges realistic computational modeling[12]. Previous network models using ionic conductance-based neurons have been developed only for the granular layer[13,14]. The only model encompassing the granular and molecular layer altogether made use of single-point neurons with a simplified representation of membrane excitability[15]. Although those models showed a remarkable predictive power against specific target parameters, their main limitation was that connectivity was set independently from neuronal morphology[13–15] preventing a direct link between microcircuit structure, function and dynamics. In the meanwhile, detailed computational models of the main cerebellar cortical neurons, which were based on morphological reconstructions embedding multiple membrane ionic channels and synaptic receptors, have been developed, tested and validated[16–20]. Thus, with the BSB, we have been able to generate the first computational model of the entire cerebellar cortical microcircuit including both the granular and molecular layer, in which multicompartmental neuron models were wired through a connectome defined by the anisotropy of dendritic and axonal processes through principled rules. The model allowed then to simulate network dynamics and validate it against naturalistic inputs[21–23].

This work generates de facto a new model-based ground truth for the cerebellar cortical microcircuit, predicting the weight that some connections should have to balance the internal activity. On the scale used here, we observed a set of emerging spatio-temporal dynamics. First, background mossy fiber bombardment induced coherent oscillations throughout the granular layer under gap-junction control. Secondly, collimated mossy fibre bursts mimicking punctuate sensory stimulation generated dense clusters of granule cell activity that propagated vertically invading the overlaying molecular layer, where inhibitory interneurons controlled the emission of burst-pause patterns from Purkinje cells. Finally, synaptic changes mimicked the long-term plasticity of neuronal discharge observed during cerebellar learning. Thus, simulations unveil local microcircuit computations explaining the neural correlates of behaviour, suggesting that the BSB cerebellar model provides a valid resource for future experimental and theoretical investigations.

## Results

### The Brain Scaffold Builder (BSB).
Cerebellar modelling using realistic morphologies poses specific problems, mostly related to the anisotropy and regular geometry of the network, that are not easily manageable with existing modeling tools[8–10] so that we developed the BSB, the first component framework (i.e., a set of well-defined interfaces that establish the protocols for component cooperation within the framework) for multiscale neural circuit modeling. The BSB allowed to easily solve construction problems like the precise orientation of neuronal processes in the 3D space, the connectivity of neurons through prescribed rules dictated by anatomo-physiological measurements and the choice of a variety of intersection rules depending on network geometry. The BSB operated through a sequence of independent steps: network configuration, reconstruction and simulation (Fig. 1a). The network volume was defined first along with cell types, then the BSB proceeded with cell placement and connectivity, reconstructing the microcircuit network (Fig. 1b, c). Finally, the BSB was interfaced with the NEURON simulator and network activity was simulated and the results visualized. Details on BSB operations are given in the Methods and Supplemental Material.

### Cerebellar network reconstruction.
The BSB was applied to the mouse cerebellar cortical network, which has a geometrically organized architecture that has been suggested to imply its computational properties[11,12]. The reconstruction and simulation of a network volume of $17.7 \cdot 10^{-3}$ mm$^3$ is reported, including the following cell and fiber types: mossy fiber (mf), glomerulus (glom), granule cell (GrC) with ascending axon (aa) and parallel fiber (pf), Golgi cell (GoC), Purkinje cell (PC), and molecular layer interneurons (MLI) comprising stellate cells (SC) and basket cells (BC).

*Neuron placement.* The network elements summed up to 29˙230 neurons (GrC, GoC, PC, SC, BC) plus 2˙453 other elements (mf, glom), which were placed in the network volume according to anatomical data[12,13,24] (Fig. 2a). The density values matched the targets given in the configuration file, the nearest neighbour and the pairwise distance distribution always exceeded cell diameter, and radial distribution function demonstrated the homogeneity of cell distribution without overlapping (Fig. S1).

*Neuron connectivity.* The network connections summed up to 1˙500˙000 chemical synapses and 2˙100 electrical synapses. The cerebellar connectome was modelled combining probabilistic and geometric rules that were chosen depending on available data and the nature of fiber (axon and dendrites) crossing (Fig. 2b–d; see Methods for details). This flexible management of connection rules is unique and fixes problems not easy to solve with cerebral cortex simulators, which deal with isotropic cellular organizations and adopt a limited number of intersection rules for all neurons and connections[8–10].

The well-known connectivity of mf and glom was entirely accounted for by literature data. The BSB generated local anisotropic glom clusters extending 60 μm along the x-axis and 20 μm along the z-axis[25], with ~20 gloms per mf[26]. Imposing that each GrC sends its 4 dendrites to gloms belonging to different mfs within about 30 μm, the BSB yielded 49 GrCs per glom on average[27,28]. Each of the 4 GrC dendrites, in addition to a single excitatory synapse on the terminal compartment, also hosted 1 inhibitory synapse on the preterminal compartment, mostly originating from different GoCs (Fig. 2c)[29,30].

The connectivity of GoCs was faced using either literature data (glom-GoC) or adopting various intersection rules (aa-GoC, pf-GoC, GoC-GoC). In fair agreement with literature, each GoC received excitation from 56 different gloms and each glom collected basolateral dendrites from ~2 GoCs[31]. There were 320 aa synapses on basolateral dendrites and 910 pf synapses on apical dendrites per GoC, all from different GrCs (Fig. 2b)[32].

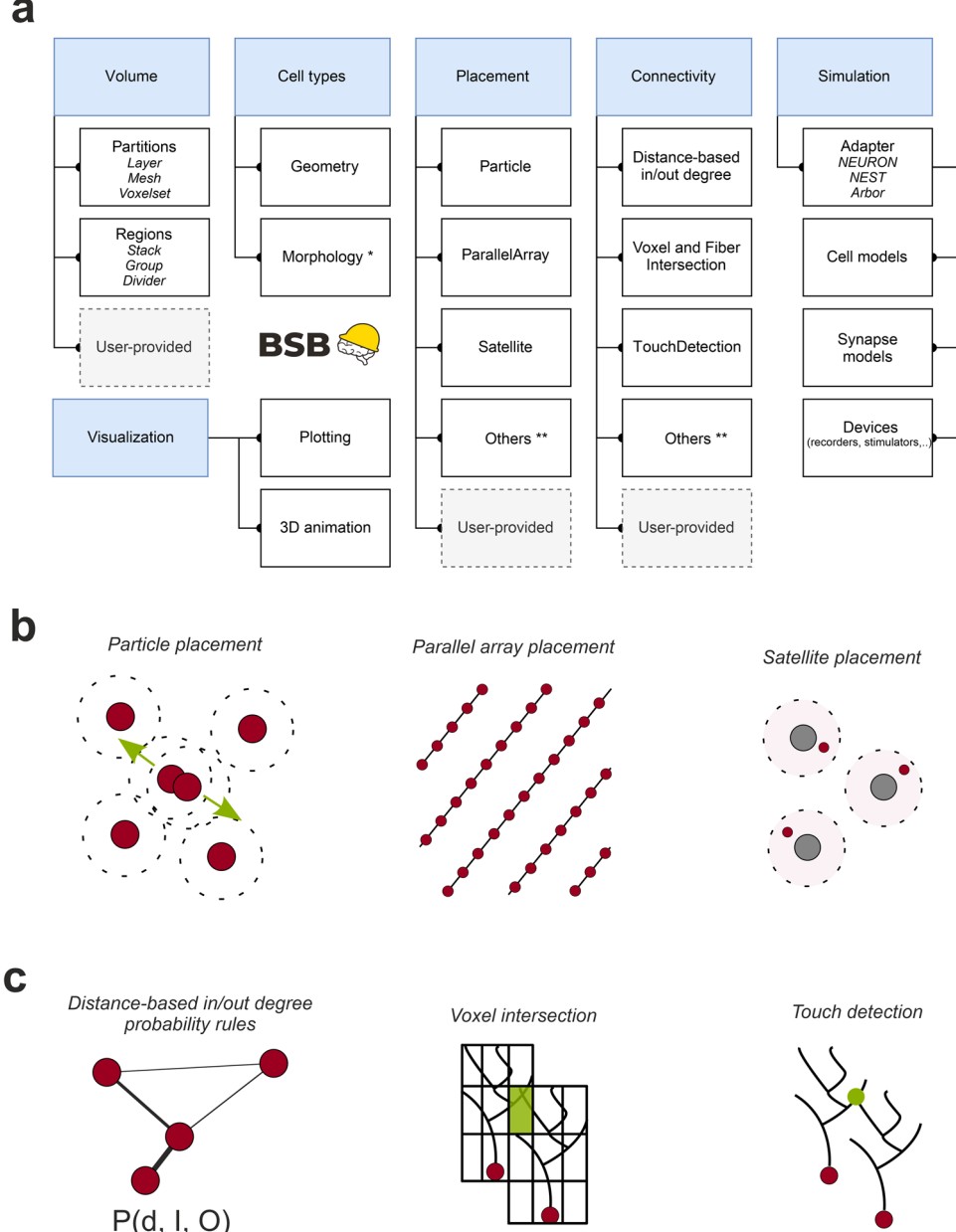

**Fig. 1 The Brain Scaffold Builder. a** Core BSB operations. In the reconstruction phase, BSB proceeds by sequentially defining the network *volume*, *cell types*, *cell placement*, *cell connectivity*. Once neurons and fibers are positioned, their geometries/morphologies are imported, and connection rules allow to wire them up and to build the network connectome. In the simulation phase, *neuron* and *synapse models* are linked to simulators, like NEURON in the present case, by a specific adapter and interfaced to a set of *devices* for stimulation and recording. In the post-simulation phase, graphic tools are made available for data representation. This workflow is applicable to any kind of brain neuronal network. **b** Infographic representations of the main *placement strategies* available in BSB, using kd-tree partitioning of the 3D space (particle placement, parallel array placement, satellite placement). **c** Infographic representations of the main *connection strategies* available in BSB: distance-based in/out degree probability functions, voxel (or fiber) intersection based on voxelization of morphologies, touch detection.

Moreover, each *GoC* received inhibition from 16 other *GoCs*[33] on basolateral dendrites (subsequent functional calibration implied ~160 synapses per pair, see below). Finally, there were ~8 *GoCs* that formed gap junctions on other *GoCs*, with ~3.5 gap junctions per pair[34].

The connectivity of *PCs* and *MLIs* was recovered using suitable intersection rules (*aa-PC*, *pf-PC*, and all *MLI* synapses). The BSB identified 1·500 *pf* synapses per *PC* (this figure was limited by the 200-μm network size along *z*-axis but it would range up by 1 order of magnitude in an unbounded volume[18,35]) and 197 *aa* synapses per *PC* from 82 different *GrCs*[36]. There were 480 *pf* synapses per *SC* and 740 *pf* synapses per *BC*, while *MLI* reciprocal inhibition[37] involved 14 *SC-SC* and 14 *BC-BC* connections with ~100 synapses per pair. The *SC* axon, mainly extending on the coronal plane, innervated ~2 *PCs*[38] and each *PC* received synapses from ~5 *SCs* (Fig. S2). The *BC* axon, mainly extending on the sagittal plane, innervated ~14 *PCs* and each *PC* received synapses from ~20 *BCs* (akin with the figure of 3-50 baskets around the *PC* soma and 7-10 *PCs* per *BC*)[38,39]. These predictions of structural parameters were further assessed and tuned through functional simulations (see below).

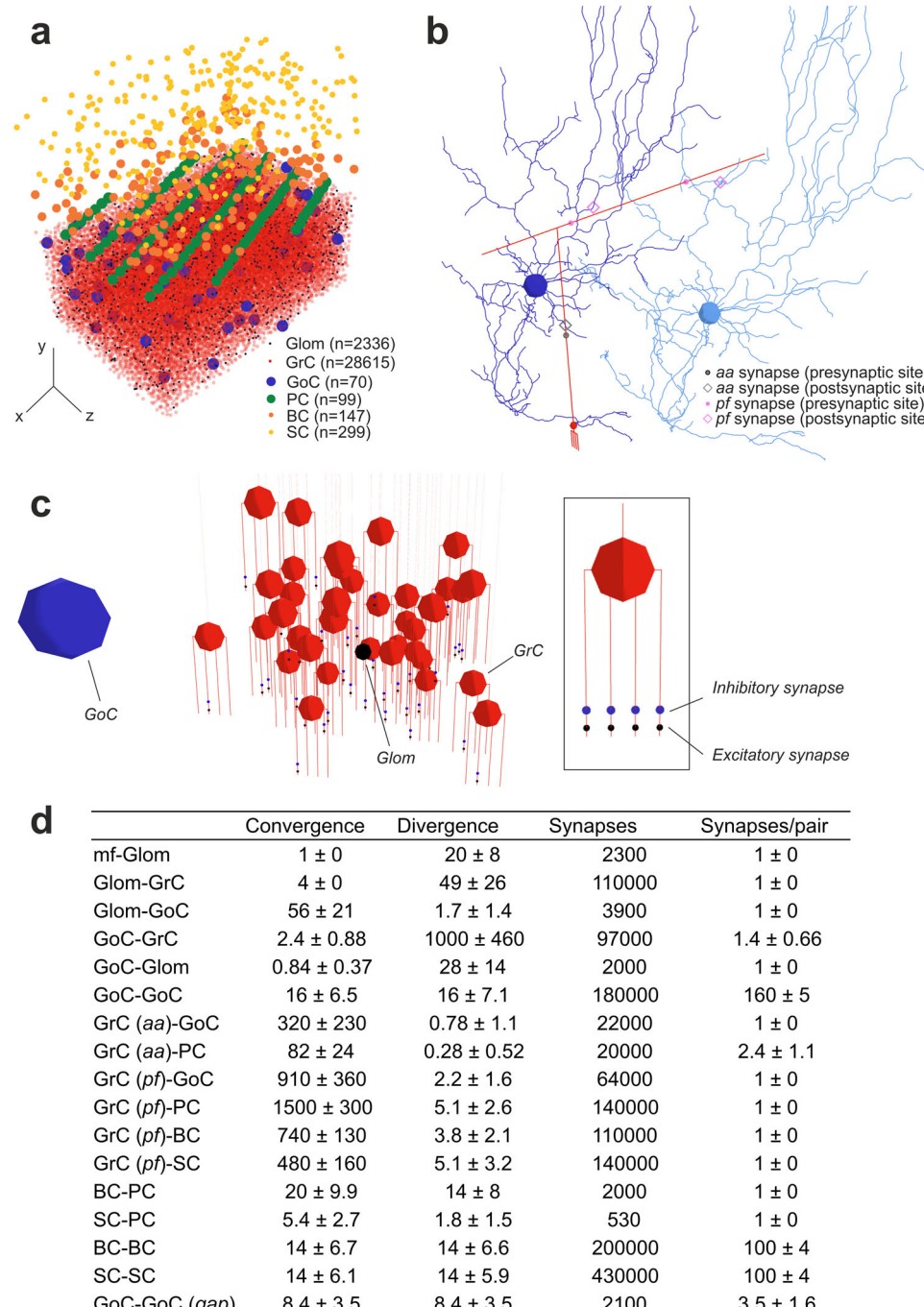

| | Convergence | Divergence | Synapses | Synapses/pair |
|---|---|---|---|---|
| mf-Glom | 1 ± 0 | 20 ± 8 | 2300 | 1 ± 0 |
| Glom-GrC | 4 ± 0 | 49 ± 26 | 110000 | 1 ± 0 |
| Glom-GoC | 56 ± 21 | 1.7 ± 1.4 | 3900 | 1 ± 0 |
| GoC-GrC | 2.4 ± 0.88 | 1000 ± 460 | 97000 | 1.4 ± 0.66 |
| GoC-Glom | 0.84 ± 0.37 | 28 ± 14 | 2000 | 1 ± 0 |
| GoC-GoC | 16 ± 6.5 | 16 ± 7.1 | 180000 | 160 ± 5 |
| GrC (aa)-GoC | 320 ± 230 | 0.78 ± 1.1 | 22000 | 1 ± 0 |
| GrC (aa)-PC | 82 ± 24 | 0.28 ± 0.52 | 20000 | 2.4 ± 1.1 |
| GrC (pf)-GoC | 910 ± 360 | 2.2 ± 1.6 | 64000 | 1 ± 0 |
| GrC (pf)-PC | 1500 ± 300 | 5.1 ± 2.6 | 140000 | 1 ± 0 |
| GrC (pf)-BC | 740 ± 130 | 3.8 ± 2.1 | 110000 | 1 ± 0 |
| GrC (pf)-SC | 480 ± 160 | 5.1 ± 3.2 | 140000 | 1 ± 0 |
| BC-PC | 20 ± 9.9 | 14 ± 8 | 2000 | 1 ± 0 |
| SC-PC | 5.4 ± 2.7 | 1.8 ± 1.5 | 530 | 1 ± 0 |
| BC-BC | 14 ± 6.7 | 14 ± 6.6 | 200000 | 100 ± 4 |
| SC-SC | 14 ± 6.1 | 14 ± 5.9 | 430000 | 100 ± 4 |
| GoC-GoC (gap) | 8.4 ± 3.5 | 8.4 ± 3.5 | 2100 | 3.5 ± 1.6 |

**Fig. 2 Reconstruction of the microcircuit of cerebellar cortex. a** Positioning of cell bodies in a 3D slab (300 × 295 x 200 μm³) of mouse cerebellar cortex. Cell numbers are indicated (the symbols reflect soma size). In this and the following figures, the *xyz* reference system is defined by *x-y* (sagittal plane), *x-z* (horizontal plane), *z-y* (coronal plane), as in standard anatomical representation. Thus, *y* measures cortex thickness (*aa* direction), while *z* identifies the major lamellar axis (*pf* direction). **b** Example of 3D morphologies illustrating *GrC-GoC* connections through *aa* and *pf*. One *GrC* and two *GoC*s are shown: the synapse along *aa* is identified by *touch detection*, while synapses along *pf* are identified by *fiber intersection*. **c** glom-GrC and GoC-GrC connections. A *glom* contacts a group of 38 *GrC*s forming an excitatory synapse on the terminal compartment of 1 of their 4 dendrites. The *glom*, in turn, is contacted by a *GoC* nearby, which forms an inhibitory synapse on the preterminal dendritic compartment of the same *GrC*s. The inset shows a *GrC* with 1 excitatory synapse and 1 inhibitory synapse on each dendrite. **d** The cerebellar cortical connectome generated by BSB reporting convergence (on the postsynaptic element), divergence (from the presynaptic element), total number of synapses, and number of synapses for each connected pair. It should be noted that *mf-glom* is not a proper synapse but just a branching.

**Cerebellar network simulations**. Network simulations were carried out using detailed neuronal and synaptic models written in NEURON for *GrC*[17], *GoC*[16], *PC*[18,20], *SC* and *BC*[19]. Local microcircuit responses to input patterns were tracked back to individual neurons and used to follow signal propagation with unprecedented resolution. All simulations were carried out in the presence of background noise to improve comparison with recordings in vivo. The emerging spatio-temporal dynamics provided functional model validation beyond constructive validity based on internal connectivity and single neuron responses (Movie S1).

**Resting state activity of the cerebellar network.** A random input at low frequency (4 Hz Poisson) on all $mfs$[22] was used to simulate the cerebellar network in resting state in vivo. Since anatomical data about the connectivity of cerebellar neurons are incomplete, but their resting discharge frequency is known, we finetuned the number of connections per pair against target values of basal discharge. The turning point was to calibrate $GoC$-$GoC$ inhibition, which influenced resting state activity of the entire network. Since the synaptic conductance (~ 3200 pS) and the number of interconnected $GoC$s (about 15) are known[33], we tuned the number of $GoC$-$GoC$ synapses until basal discharge frequency was achieved. Eventually, the background frequency of all cerebellar neuron types fell in the ranges reported in vivo in anaesthetized rodents ($mfs$: 4.2 ± 2.6 Hz; $GrC$s: 0.81 ± 1.3 Hz; $GoC$s: 19 ± 15 Hz; $PC$s: 31 ± 1.6 Hz; $BC$s: 11 ± 5.1 Hz; $SC$s: 9.4 ± 12 Hz) [$GrC$s[40], $GoC$s,[16,41,42] $PC$s[43], $SC$s and $BC$s[44–46]].

**Granular layer oscillations and synchrony.** Background $mf$ activity is known to generate synchronous low-frequency oscillations in the granular layer[47]. Indeed, in the model, the FFT of $GoC$ and $GrC$ firing revealed a synchronous oscillatory behaviour in the theta band, with the first harmonic peaking at 9.7 Hz. When $GoC$-$GoC$ gap junctions were disabled, the regularity of the oscillation decreased and the first FFT harmonic moved out of theta band (Fig. 3a)[48].

To investigate the sensitivity of Golgi cell synchrony to gap junction density[49], we compared the cross-correlation of Golgi cell discharge with the degree of coupling (electrotonic distance, Fig. S3) in GoC pairs, when the network was activated with 4-Hz Poisson mossy fibre activity. The cross-correlation of Golgi cell discharge decreased smoothly with the increase of electrotonic distance (Fig. 4a), tending toward a non-zero level. This non-zero level, that indicates the vanishing of gap-junction effects, corresponded to that observed by disabling the gap junctions and unveiled the synchronizing effect of the feedback loops passing through the granule cell – Golgi cell circuit reported earlier[14,50] (see below). (Fig. 4a). This loose synchronization due to shared input from GrCs was still correlated to spatial proximity. In Golgi cell pairs with direct coupling ($n = 384$ out of 4830 pairs), increasing the gap-junction density by 2.5 times caused two discrete peaks (at −1ms and + 1ms) in the mean cross-correlogram (Fig. 4b). A spike could either precede or follow the one emitted by a neighbouring Golgi cell with millisecond precision as observed experimentally[49]. In Golgi cell pairs with indirect coupling (i.e., 2 or more cells away, $n = 842$ pairs), the two peaks in the mean cross-correlogram disappeared, as much as when gap junctions were disabled. The percentage of synchronous spikes across all GoC pairs located within 100 μm reached about 27% with a 5-ms time lag window, again consistent with experimental findings[49] (Fig. 4c). Following this functional validation, the model was used to compute the probability density of spike coincidence in the granular layer, predicting that the effects of Golgi cell coupling can extend over an ellipsoidal volume over ~100 × 200 μm.

**Impulsive response of the cerebellar network.** Short stimulus bursts were delivered to a bundle of 4 $mf$s connected to ~80 $glom$s to emulate whisker/facial sensory stimulation in vivo[22,40]. The burst propagated through the network, temporarily raising neuronal firing (Fig. 3b, Movie S1). The relationship between the number of spikes at afferent synapses and the response frequency to the $mf$ burst was robustly captured by multiple linear regression (Fig. 3c; Fig. S4,a; Table S1).

GrC responses: Fundamental predictions on how $GrC$s respond to incoming bursts derive from current clamp recordings in situ[51] and simulations[17], which revealed the role of synaptic receptors

and ionic channels. In BSB simulations, bursts on a collimated $mf$ bundle activated a dense cluster of $GrC$s[15,21,52]. The relationship between the number of input spikes (both at $GoC$-$GrC$ and $glom$-$GrC$ synapses) and $GrC$ response frequency unveiled 4 groups of $GrC$s with a corresponding number of synaptically activated dendrites (Fig. 3c). The number of $GrC$ spikes, first spike latency and dendritic $[Ca^{2+}]_{in}$ correlated with the number of active dendrites (NMI = 0.71, 0.86, 0.59, respectively) (Fig. 5a, b).

When the inhibitory mechanisms (comprising transient and persistent inhibition) were disabled to simulate a pharmacological $GABA_A$ receptor blockade, (i) $GrC$ baseline frequency increased, (ii) a tail discharge appeared after the burst, (iii) responses including more spikes appeared, (iv) the first spike latency decreased, and (v) response variability decreased (Fig. 5a, b). The number of $GrC$ spikes, first spike latency and dendritic $[Ca^{2+}]_{in}$ still correlated with the number of active dendrites (NMI = 0.79, 0.85, 0.61, respectively) (Fig. 5b). Interestingly, inhibition caused a reduction in the number of active $GrC$s (i.e. those firing >= 1 spike in the 40 ms after the $mf$ burst onset were 3390 ± 431, and 8348 ± 1724 with GABA-A off; $n = 10$ simulations; $p < 0.001$, unpaired $t$-test) but enriched the spike pattern, as predicted theoretically[11,53].

Recordings in vivo disclosed precise integration of quanta and high-fidelity transmission in the granular layer[22,54–57]. In BSB simulations, $GrC$s receiving maximum excitation generated one action potential for each spike of the input burst, with short latency (<2 ms), and faithfully followed the input up to 250 Hz (Fig. 5a) (Movie S2).

GoC responses: Following punctuate sensory stimulation in vivo, $GoC$s have been reported to respond with short bursts of 2–3 spikes at up to 200–300 Hz[58]. In BSB simulations, $GoC$s immersed in the $GrC$ active cluster generated a burst of 2-5 spikes with a maximum instantaneous frequency of 213 ± 29 Hz (Fig. 5c). When GABA synapses and gap junctions between $GoC$s were disabled, the response bursts showed up to 6 spikes, with a higher maximum instantaneous frequency (308 ± 16 Hz) ($n = 70$ $GoC$s; $p < 0.001$, paired $t$-test) (Fig. 5c). The burst was caused by synaptic excitation relayed by $glom$s and $GrC$s (through both $aa$s and $pf$s), which generated AMPA and NMDA currents in $GoC$ dendrites (Movie S3). The "silent pause" appearing after the burst was caused both by an intrinsic phase-reset mechanism[58–60] and by reciprocal inhibition between $GoC$s, demonstrating marked dendritic processing capabilities[16].

PC and MLI responses: $PC$s in vivo are known to respond to punctuate stimulation with burst-pause patterns[23,61]. In BSB simulations, $PC$ responses depended on cell position relative to the $mf$ active bundle (Fig. 6a). The $PC$s placed vertically on top of the $GrC$ active cluster received the largest number of $aa$ and $pf$ synaptic inputs producing typical burst-pause patterns[18]. The $burst$ $coefficient$ was correlated with the number of synaptic inputs from $pf$ and $aa$ (multiple regression analysis: $R^2 = 0.91$) (Fig. 6b). The $pause$ $coefficient$ was correlated with the $burst$ $coefficient$ (NMI = 0.79) and with the number of spikes from $MLI$s (NMI = 0.66) (Fig. 6b), reflecting the origin of the pause from both intrinsic after-hyperpolarizing mechanisms and $MLI$ inhibition[62]. Indeed, $MLI$s are known to narrow the time window and reduce the intensity of $PC$ responses[44]. In BSB simulations, the $PC$ AMPA current arose soon after the spikes emitted by $GrC$s, while the $PC$ GABA current was delayed by 2.6 ms (Fig. 6c). In summary, the di-synaptic IPSCs produced by $MLI$s quickly counteracted the monosynaptic EPSCs produced by $aa$s and $pf$s, providing precise time control over $PC$ activation[52,63].

$BC$s in vivo are known to generate lateral inhibition reducing $PC$ discharge below baseline causing contrast enhancement[44,53].

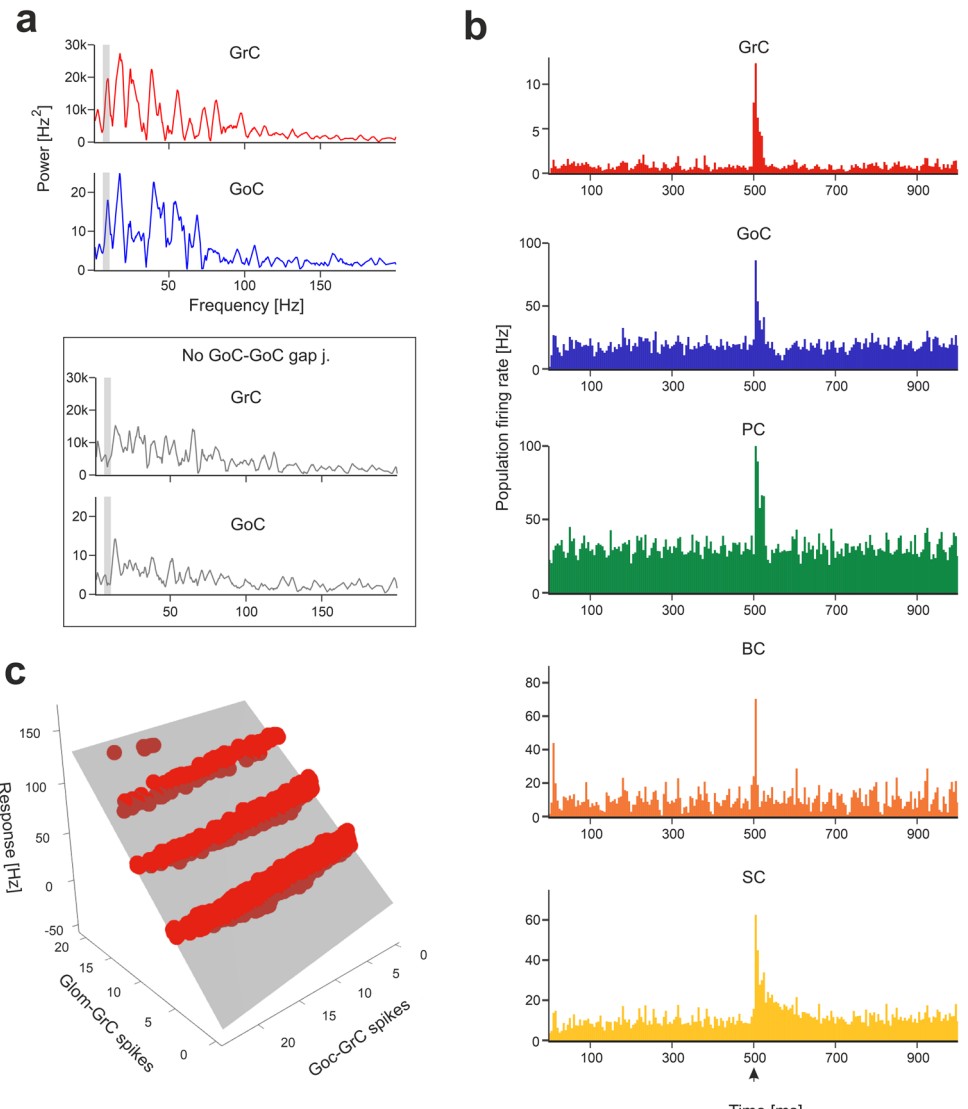

**Fig. 3 Network responses to background noise and *mf* bursts. a** Power spectra of *GrC* and *GoC* activity are computed with Fast Fourier Transform (FFT) of spike time series (total population spike-counts in 2.5 ms time-bins). The periodicity of peaks in power spectra reveals synchronous low-frequency oscillations in the granular layer. The grey curves represent the power spectra when *GoC-GoC* gap junctions were disabled, showing a marked decrease in periodicity. The grey bands correspond to mouse theta-band (5-10 Hz). **b** The Peri-Stimulus-Time-Histograms (PSTH) of each neuronal population show the effect of the localized *mf* burst (onset indicated by arrowhead) emerging over background noise. The PSTHs show number of spikes/5 ms time-bins normalized by the number of cells, averaged over 10 simulations. **c** Example of multiple linear regression of *GrC* responses (firing rate) against the number of synaptic spikes from *gloms* and *GoC*s, during 40 ms after stimulus onset. The grey surface is the fitted plane to the points (each point corresponds to a *GrC* receiving the *mf* burst on at least 1 dendrite).

In BSB simulations, this pattern emerged during stimulation of a *mf* bundle (100 ms @ 50 Hz stimulation on 24 neighboring *mf*s). The *PC*s placed in a band 150-200 μm beside the active cluster along the *x*-axis were inhibited, bringing their frequency below baseline. When *MLI-PC* synapses were disabled, the effect disappeared revealing contrast enhancement due to lateral inhibition (Fig. 6d).

The response of *MLI*s in vivo is only partially known[53]. In BSB simulations, *SC*s and *BC*s intersected by active *pf*s responded to input bursts and their activity remained higher than baseline for several hundreds of milliseconds, especially in *SC*s[19] (Fig. S4,b).

*Modification of model parameters to simulate neural correlates of behavior.* Two conditions modifying PC firing patterns and their modulation were explored in order to test whether our network model was able to predict neural correlates of behavior: i) knock-out

(KO) of MLI inhibition on PCs, which impacts on vestibulo-ocular reflex (VOR) adaptation[64], and ii) long-term plasticity at *pf*- PC synapses, which drives learning in associative tasks like eye-blink classical conditioning (EBCC)[65].

KO of MLI inhibition on PCs: GABA$_A$ receptor–mediated synaptic inhibition was selectively disabled in Purkinje cells (KO condition), and a single stimulus pulse was delivered to a bundle of 13 *mf*s. The burst response of PCs was broader and with a strong temporal dispersion (jitter) of simple spikes in KO than control condition (control: 0.49 ms; KO: 1.01 ms; $p < 0.01$ t-test) (Fig. 7). These alterations of PC activity patterns reproduced the dysregulation of cerebellar signal coding and adaptation observed in *PC-Δγ2*, a mouse line in which GABA$_A$ receptor-mediated synaptic inhibition was selectively knocked-out in Purkinje cells[64].

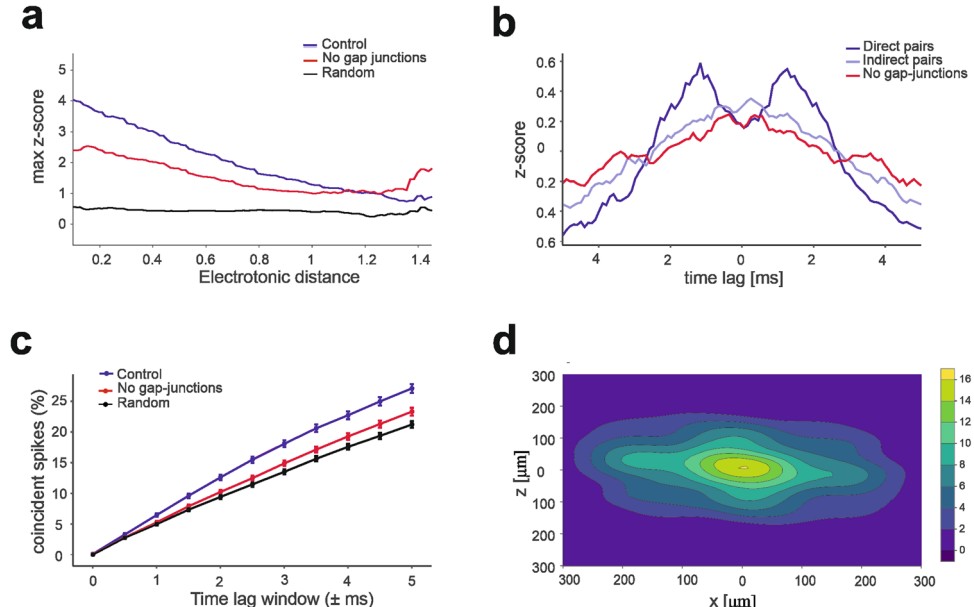

**Fig. 4 *GoC* millisecond synchronization by gap junctions. a** Maximum cross-correlation in pairs of Golgi cells as a function of electrotonic distance. The three curves represent control condition (4-Hz Poisson mossy fibre activity), with gap-junctions disabled, and with random spike patterns of *GoC*s. All values were calculated using a sliding window of $+ -0.2$ electrotonic distance. At large electronic distances, the z-score in control conditions tends toward the value set by random input patterns. **b** The average cross-correlograms (0.5 ms bins) is calculated in control condition for *GoC* pairs at <100 μm distance with either direct coupling ($n = 384$), indirect coupling ($n = 842$), all pairs located <100 μm distance from each other when gap junctions were disabled when gap-junctions are disabled. The z-score shows two distinct peaks indicating *GoC*-*GoC* correlation with ms spike precision with on average 7.5 gap-junctions per direct pair. **c** The percentage of spikes that fall within distinct time-lag windows across all pairs located <100 μm distance in control condition, with gap-junctions disabled, and with random spike patterns of *GoC*s. Points are mean ± SEM ($n = 1181$). **d** Probability density of spike coincidence in the granular layer horizontal plane. This plot indicates that, with a *GoC* spike in [0,0], there is a certain probability that *GoC*s around it will fire a spike within a ± 5 ms time-window. The integral of the probability density function over the whole network corresponds to the average spike coincidence for the same time window in (**c**).

Long-term plasticity at pf-PC synapses: Reduced values of the AMPA receptor-mediated synaptic conductance ($g_{syn}$) were used to simulate long-term depression (LTD) at *pf*-PC synapses. In EBCC, a level of suppression of about 15% was found to correlate with a stable generation of associative blink responses at the end of the learning process[65]. In BSB simulations using a stimulus at 50 Hz on a *mf*-bundle, different LTD levels caused a corresponding amount of PC simple spike suppression (Fig. 8). A 15% PC simple spike suppression emerged with *pf*-PC LTD of about 35%, predicting the number of synapses that should undergo LTD in order to explain the experimental observation.

**Discussion**
This work shows the first detailed model reconstruction and simulations of the cerebellar cortical network and predicts neuronal activities involved in the propagation of mossy fiber input signals from the granular to the PC and molecular layer. By means of the BSB model, we have combined heterogenous data using suitable placement and connectivity rules with accurate multi-compartmental neuron models. In the optimization process, the model extracted information from the interdependence of parameters, bound at high-level through ensuing network dynamics, allowing us to fill gaps in knowledge through constructive rules. In the validation process, the model demonstrated its compatibility with a wealth of experimental literature data collected over the last decades and a parameters sensitivity able to uncover the neural correlates of specific physio-pathological conditions.

**A model-based ground-truth for the cerebellar cortical network**. The statistical and geometrical rules derived from

anatomical and physiological works[12,26] almost completely anticipated network connectivity at the cerebellar input stage. In the BSB model, each *glom* hosted ~50 excitatory and ~50 inhibitory synapses on as many *GrC* dendrites, plus ~2 excitatory synapse on basolateral dendrites of as many *GoC*s, summing up to ~102 synapses per *glom*, in agreement with the anatomical upper limit of ~200[30]. Each one of the 4 *GrC* dendrites received an excitatory and (in most cases) an inhibitory input from as many different *mf*s and *GoC*s, respectively[29,31]. Each *GoC* received ~320 *aa* synapses on basolateral dendrites and ~910 *pf* synapses on apical dendrites, according to the figure of ~400 and ~1200[32], and there were ~3 electrical synapses per *GoC*-*GoC* pair[48]. Functional tuning suggested that the number of gap-junction could actually be 2.5 time higher, i.e., ~7-8 per *GoC*-*GoC* pair[48]. Only the number of *GoC*-*GoC* GABAergic synapses, which amounted to a figure of 160 after functional tuning, lacked any experimental counterpart. In the molecular layer, under geometric and functional constraints, the BSB model placed limits to the debated numbers determining *PC* and *MLI* connectivity. The model predicted that ~25% of *aa*s contacted the distal dendrites of the overlaying *PC*s (7·133 out of 28·615 *GrC*s), each *aa* forming 2.4 synapses on average, supporting the important role predicted for the *aa*[63,66], while *pf*s formed 1 synapse per *PC* dendritic intersection. In summary, each *PC* received 12% of the whole *GrC* inputs from *aa*s, matching the empirical estimate of 7-24%[67]. The BSB generated ~25 SC-PC and BC-PC synapses altogether, which compares well with the experimental estimate of ~20[68]. Moreover, there were ~17·600 *pf*-MLI-PC synapses (~2·600 *pf*-SC-PC and ~15·000 *pf*-BC-PC synapses), compatible with the prediction that the *pf*-MLIs-PC input is larger

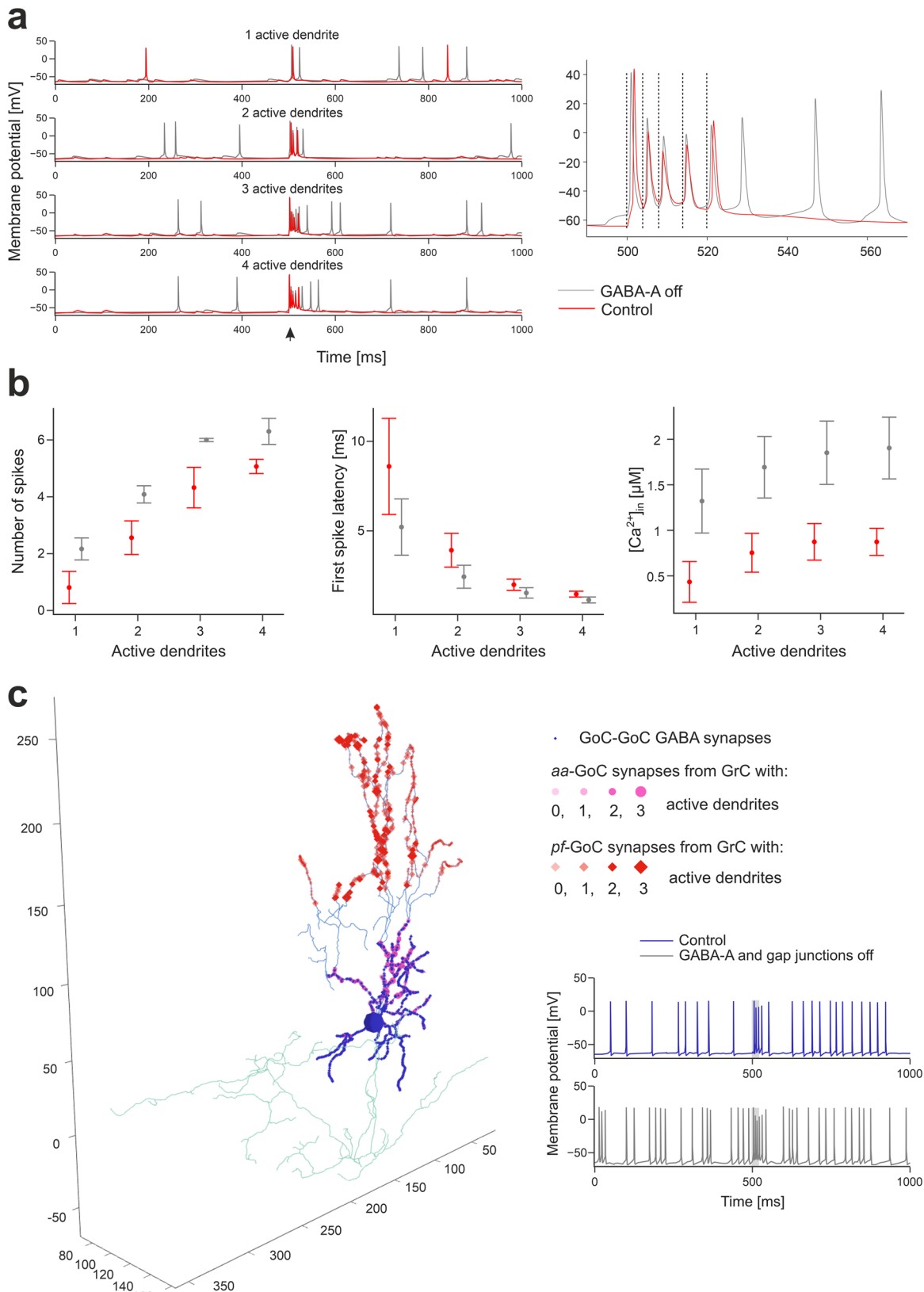

than the pf-PC input on the same PC[69]. In general, since all dendritic trees in the molecular layer are orthogonal to pfs, the BSB reconstruction ranked the number of synapses according to dendritic size - PC (~1'500) > GoC (~900) > BC (~700) > SC (~500) – a figure that would increase proportionately by scaling the model slab to include full-length pfs[70].

Accurate single-neuron models with realistic morphology proved also critical to carry out simulations allowing us to finetune the connectome. In particular, the number of inhibitory synapses per GoC-GoC pair was increased in order to make them fire at ~19 Hz [2-30 Hz range[40,58]]. Similarly, the number of inhibitory synapses per SC-SC and per BC-BC pairs was tuned in

**Fig. 5 Granular layer activation. a** Membrane potential of 4 representative *GrCs* with 1 to 4 dendrites activated by the *mf* burst (20 ms@200 Hz over background noise, onset indicated by arrowhead), in control condition and after GABA-A receptors blockade ("GABA-A off"). The burst response of the *GrC* with 4 active dendrites is enlarged on the right to highlight spike-timing (dashed lines indicate the *mf* burst spikes). **b** Number of spikes (measured in the 40 ms from *mf* burst onset), first spike latency, and dendritic $[Ca^{2+}]_{in}$ (measured in the 500 ms from *mf* burst onset) in subgroups of *GrCs* with the same number of activated dendrites ($N_{dend}$). Means ± sd are reported (n = 21068 with $N_{dend}$ = 0, n = 2361 with $N_{dend}$ = 1, $n$ = 892 with $N_{dend}$ = 2, n = 164 with $N_{dend}$ = 3, $n$ = 6 with $N_{dend}$ = 4). The graphs compare responses in control and during "GABA-A off". **c** Synapses of a *GoC* activated by *GrCs*. Bigger markers correspond to presynaptic *GrCs* more activated by the *mf* burst. The GABAergic synapses from other *GoCs* are on basolateral dendrites, *aa* synapses are on basolateral dendrites, *pf* synapses are on apical dendrites. In this example, the *GoC* receives 30% of its *aa* synapses and 6% of its *pf* synapses from *GrCs* with at least 2 active dendrites. Traces on the right show the *GoC* membrane potential in response to the *mf* burst (same stimulation as in (**a**), grey band) in control and during GABA-A receptors and gap junctions switch-off.

order to make them fire at ~10 Hz [1–35 Hz range[44,45]] and to bring *PCs* into their resting state frequency range of ~31 Hz [36.4 ± 11.5 Hz[43,71]] in vivo. The number of gap-junctions per *GoC-GoC* pair was tuned to obtained millisecond synchrony[49].

Thus, a reconstruction of model connectivity purely based on geometrical rules was not sufficient and a careful tuning against functional data was needed. This two-pronged (structural and functional) approach ensured that all parameters were bound at high-level through the basal neuronal firing frequency at rest in vivo[1]. Eventually, the network connectome is in fair agreement with a wealth of disparate anatomical and functional determinations, suggesting that the emerging picture provides a new model-based ground-truth for the cerebellar cortical network.

**Cerebellar network model validation and predictive capacity.** The functional validation of single neuron models was previously reported in specific studies[16,17,19,62], so that these neurons could be directly plugged in and used to simulate spatio-temporal network dynamics in vivo. The functional validation of the cerebellar network model implied first to analyse responses to diffused background noise, which is reported to generate coherent large-scale oscillations[47]. The BSB model showed indeed that *GrCs* and *GoCs* were entrained into low-frequency coherent oscillations in resting state and, interestingly, this happened under gap junction control as observed experimentally[48]. Furthermore, the BSB model showed that Golgi cell synchronization through gap junctions occurred with millisecond precision[49]. Thus, gap junctions refined and potentiated the synchronizing effect of massive shared excitatory inputs from GrCs reported earlier[14,50]. As a whole, these simulations predict that the spatial organization of Golgi cell inhibitory control depends on the distance among GoCs and on their specific morphology and orientation supporting a modular circuit organization: a marked correlation and synchronicity can be observed within an assembly, while it tends to decrease between assemblies, indicating Golgi cells coordinate segregation and integration of activities in the granular layer of cerebellum[72].

The functional validation was extended by simulating responses to naturalistic *mf* bursts, which rapidly propagated through the *GrC-PC* neuronal chain (Fig. 9) (Movie S1). *GrCs* responded in a dense cluster[52] regulated by *GoCs* and activated soon thereafter the overlaying *PCs* and *MLIs*. In the cluster, 45% of the GrCs fired at least one spike, in agreement with results reported previously[12,52]. Not unexpectedly, *SCs* and *BCs* effectively reduced activation of *PCs* placed either along or beside the active *pfs*, respectively, generating feedforward and lateral inhibition[11,12].

**Model predictions of neural correlates of behavior.** Network simulations with the BSB cerebellar model predicted the neural correlates of behaviour in different physio-pathological conditions. First, *PCs* showed the typical burst-pause responses that are thought to correlate with cerebellar-dependent motor control[61]. These responses were seriously altered by changing *mf-GrC*

neurotransmitter release probability[17], whose effect propagated from the cerebellar input stage throughout the whole thickness of the cerebellar cortical network, suggesting a possible substrate for pattern regulation in the cerebellum[73]. Secondly, selective removal of GABA_A receptor-mediated synaptic inhibition from PCs reproduced the neuronal alterations correlated to dysfunctional VOR adaptation in the *PC-Δγ2* mouse line[64]. Thirdly, plasticity remapping predicted that LTD in 35% of *pf-PC* synapses could explain the 15% PC simple spike suppression observed during EBCC[65].

**Comparison with previous cerebellar models.** Since Marr's work[11], the cerebellum has been amongst the most intensely modelled brain microcircuits and has provided a workbench to test biophysical principles of excitability and connectivity. The incorporation of biologically realistic features into models has progressed along the last three decades, as sketched below by considering just some of the many published works.

Spiking models of the granular layer with active membrane mechanisms in neurons. The first one[50] had only the granular layer, neurons were single compartment and with generic excitable mechanisms, synapses did not have short-term plasticity. A second model used cell-specific membrane mechanisms and synapses with short-term plasticity[14]. However, neurons did not have realistic multicompartment morphology yet. In both cases, connections respected proportions reported in literature without prescribed connectivity rules.

Spiking models including both the cerebellar cortical network and deep cerebellar nuclei. The first model[74] used integrate and fire point neurons and a canonical formulation of neuronal numbers and connectivity. A second set of models used a cerebellar network scaffold strategy with general rules for cell positioning and connectivity based on the probability cloud algorithm[13]. An extended version[75] included deep cerebellar nuclei and the inferior olive. Neurons were single compartment with non-linear discharge properties and synapses did not have short-term plasticity.

The current model of the cerebellar cortical network integrates and extended all the previous realizations by featuring an integrated reconstruction and simulation strategy, using multi-compartment neurons with cell-specific membrane mechanisms, using synapses with intrinsic neurotransmitter release dynamics and short-term plasticity, and adopting multiple connection rules including morphology-based touch-detection and voxel-intersection. These advancements reflect into the ability of the model to capture a large set of biological properties of the network under various physio-pathological conditions.

**Limitations and future challenges.** The most relevant problem of this kind of microcircuit models is to incorporate variables that remain underconstrained. Here we have 5 cell types, 16 synaptic types and as many ranges for synaptic density. Almost all of them were carefully validated beforehand, except the BC model with its

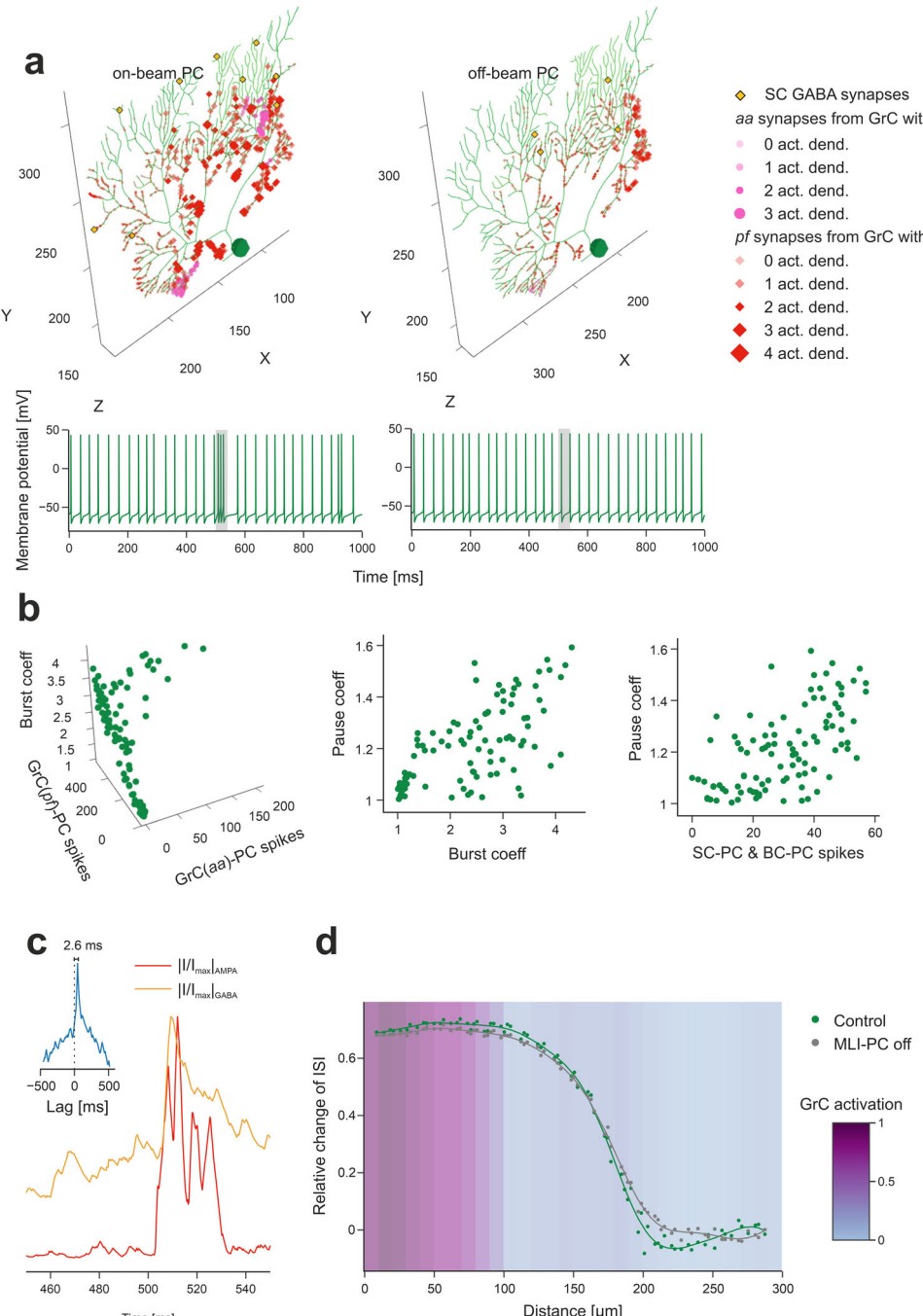

**Fig. 6 Purkinje cell activation. a** The *PC* placed on top of the *GrC* active cluster and the *PC* placed at its margin show different synaptic inputs. GABAergic synapses from *SC*s are on medium-thickness dendrites (those from *BC*s on *PC* soma are not shown), *aa* synapses are located on thin dendrites and *pf* synapses on thick dendrites. Bigger markers correspond to presynaptic *GrC*s more activated by the *mf* burst. In this example, the on-beam *PC* receives 23% of its *aa* synapses and 6% of its *pf* synapses from *GrC*s with at least 2 active dendrites, the off-beam *PC* 0% of its *aa* synapses and 0.6% of its *pf* synapses from *GrC*s with at least 2 active dendrites. The corresponding membrane potential traces are shown at the bottom (the 20 ms *mf* burst is highlighted by grey band). **b** Analysis of the burst-pause response of *PC*s to the *mf* burst (20 ms@200 Hz over background noise). The *burst coefficient* (i.e. the shortening of the inter-spike interval due to the *mf* burst, with respect to baseline) is reported against the number of spikes from *aa*s and from *pf*s (multivariate regression analysis: $R^2 = 0.91$). The *pause coefficient* (i.e. the elongation of the inter-spike interval after the *mf* burst response, with respect to baseline) is reported against either the *burst coefficient* (NMI = 0.79) or the number of spikes from *SC*s and *BC*s (NMI = 0.66). **c** Synaptic currents recorded from the *PC* on top of the *GrC* active cluster (same as in (**a**)), in voltage-clamp. The traces are the sum of all excitatory (AMPA) and inhibitory (GABA) dendritic currents during the *mf* burst. They are rectified, normalized and cross-correlated (inset) unveiling a GABA current lag of 2.6 ms with respect to AMPA current. **d** By stimulating a *mf* bundle (100 ms@50 Hz Poisson stimulation on 24 adjacent *mf*s), the *PC* response (modulation with respect to baseline) was quantified by the relative change of Inter-Spike-Interval (ISI), during the stimulus, where 0 corresponds to baseline. The two series of points compare *PC* response modulation when *SC*s and *BC*s were either connected ("control") or disconnected from *PC*s ("*MLI-PC* off"). The curves are regression fittings to the points (Kernel Ridge Regression using a radial basis pairwise function, from Python scikit-learn library). The *GrC* active cluster ("*GrC* activation") was identified by a threshold on the stimulation-induced activity by using kernel density estimation.

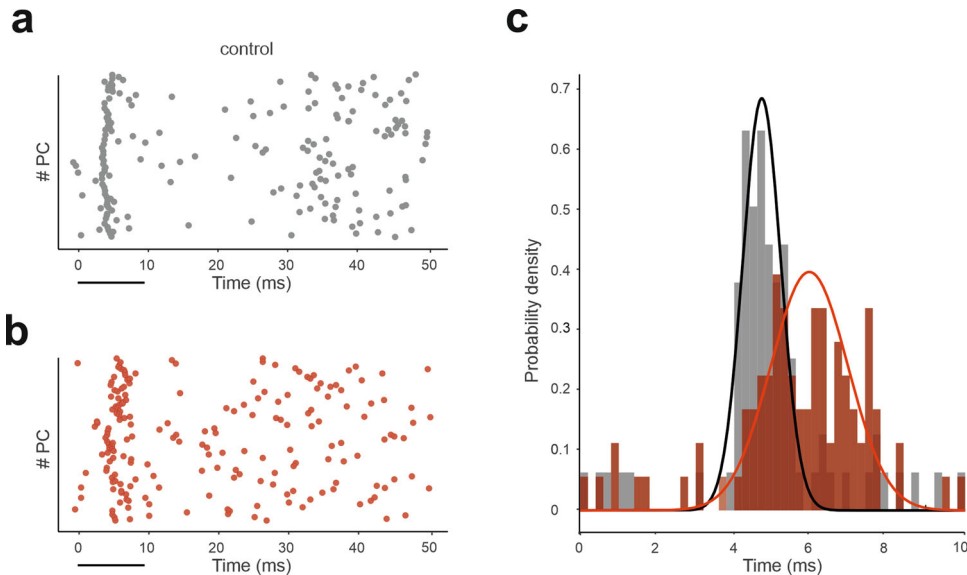

**Fig. 7 Molecular layer interneurons modulate PC discharge.** Raster plots of *PC* spikes following an impulse on a bundle of 13 mfs (time 0) in (**a**) control condition and in (**b**) KO condition, in which GABA$_A$ receptors are blocked from *PC*s to uncover the neural correlates of dysfunctional VOR adaptation in the *PC-Δγ2* KO mouse line (**c**) Probability density functions of spike count (time bins of 0.2 ms) in the 10-ms window following stimulation in the two conditions. Note the more scattered firing response in KO condition.

synapses and the density of reciprocal interneuron inhibitory synapses, which warrants specific investigation. Thus, although the parameterization of the cerebellar network model relies on one of the best-defined anatomical and physiological datasets in the brain[12–14,76], it cannot be excluded that other parameter combinations might also be effective. Indeed, some structural data were missing (e.g., not measured experimentally) or error-prone and their estimates were provided by network reconstruction and simulation. The emergent structural parameters were then confronted with available knowledge for constructive validity. The scaffold configuration allows to easily host new data and to update the existing ones when new experimental data become available.

Here we have enforced a connectivity principle largely based on proximity rules between neurites and tuned the connection algorithms to bring the connectivity within the anatomo-physiological range (see Methods). Alternative algorithms for automatic parameter tuning may also be used to predict the cerebellar cortical network connectome[77] and compared to the present results. Finally, while we have used two most representative functional templates (background oscillations and response to sensory-burst stimulation in vivo), others could be envisaged. It should be noted that often functional validation relies on sparse experimental data quantifying single-neuron responses to sensory stimuli. Therefore, multi-layer mesoscale recordings would be useful to further validate model predictions about the mechanisms of microcircuit computation in the cerebellum, e.g., following whisker stimulation or along EBCC training.

Although it is validated on a small network scale (30 k neurons and 1.5 M synapses), the model is about 1000 times smaller than the whole mouse cerebellum. This would not be a problem if the model would be a small-scale representation of the cerebellar cortex, but this is not the case given the anisotropy of cerebellar network architecture. The first issue is that signal propagation along the transverse plane would require longer modules. Here we have observed the formation of vertical columns[63,67] but it would be important now to assess[78,79] the beam hypothesis along with spatial signal filtering and plasticity[11,16–19,53]. Moreover, the cerebellar cortex is subdivided into microzones with different biochemical and functional properties, while the present model

can just be taken as a good proxy of the Z + microzone[80–82]. Therefore, the model should be extended and diversified to explore effects on a larger scale.

Another issue is that, in the model, all neurons of the same type are identical one to another. However, there is morphological and functional variability among neurons of the same type. Moreover, there are known variants of granule cells, Golgi cells and Purkinje cells[16–18,20]. It would therefore be important to explore the impact of neuronal variability, which can bring about relevant computational effect[83]. The same also applies to synapses, which now have the same release probability and gain at homologous connections but are tuned by plasticity in real life[16,21,23,35,69,76,81] and could therefore change network dynamics. The future introduction of plasticity, which now is present only in simplified models[84,85] and cerebellar subnetworks[11–15], will allow to refine the effective functional organization of the connectome and test hypotheses on network functioning.

Finally, the operations of the cerebellar cortex are tightly bound to those of the deep cerebellar nuclei and of the inferior olive. However, to date the only available representations on the mesoscale are reconstructed using simplified single point neurons[84,85] and a fine grain realistic representation is missing. Therefore, the model could be extended to the mesoscale to investigate how the cerebellar cortex operates inside the olivo-cerebellar system.

## Conclusions

In aggregate, the BSB model shows that the geometrical organization of neurites largely determines cerebellar cortical connectivity and microcircuit dynamics, supporting the original intuition of J.C. Eccles in the late 60's[11,53]. A similar conclusion was recently reported for the cortical microcolumn[4]. With appropriate extension, the model could allow to simulate cerebellar modules including differentiated micro-zones and microcomplexes[80–82] and more complex patterns of stimuli in the sensorimotor and cognitive domain[76]. Given the "scaffold" design, new neurons and mechanisms can be plugged-in to address ontogenesis, species differences (for example in humans) and pathology. For example, the model may be used to predict the emerging dynamics caused by genetic or epigenetic alterations in neuron (morphology and function) and synaptic properties, as it is

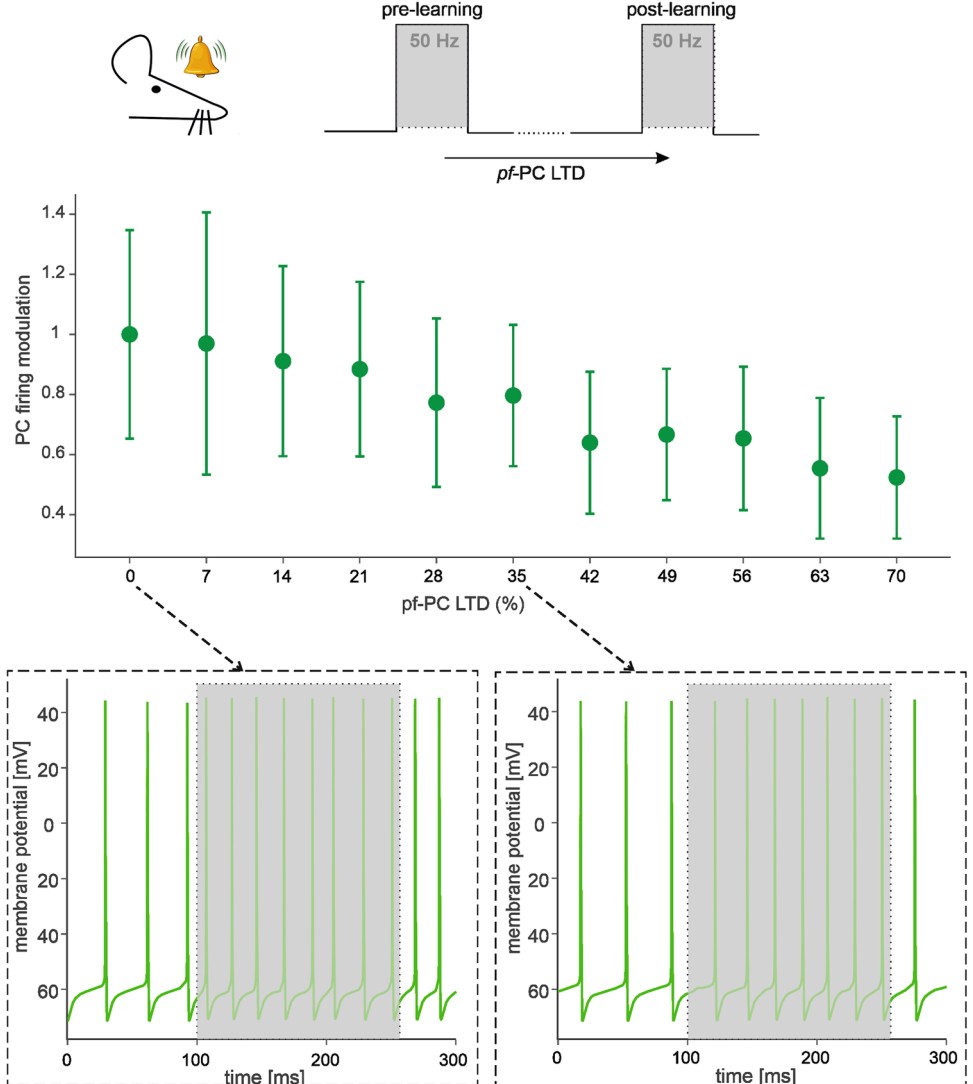

**Fig. 8 pf-PC plasticity modulates PC discharge.** The scheme on top shows the simulation protocol that emulates an EBCC paradigm, in which a conditioned stimulus (CS) is delivered to the *mf*s. Our simulations reproduce the final state ("post-learning") by exploring multiple levels of *pf-PC* LTD. The relationship between these LTD levels and PC firing modulation (relative to "pre-learning" state with 0% LTD) is shown. The points are mean ± SEM across all *PC*s (*n* = 99). Two representative traces illustrating *PC* discharge are shown for 0% LTD and 35% LTD in the insets.

supposed to happen in ataxia, dystonia and autism[86,87]. The model may also be used to predict the impact of drugs acting on ionic channels and synaptic receptors. In conclusion, the model can be regarded as a new resource for investigating the structure-function-dynamics relationships in the cerebellar network.

## Materials and Methods

**The BSB modeling framework**. The BSB is a Python package (RRID:SCR_008394, version 3.8 + ) that can be installed on any device where Python is available (*pip install bsb*) and is open source with source code documentation and topical tutorials. It includes workflows and building tools for multiscale modeling of networks (both reconstruction and simulation) and is compatible with a wide variety of target systems such as personal computers, clusters or supercomputers and provides effortless parallelization using MPI.

Effective frameworks for microcircuit modelling have recently appeared for Python such as the BMTK[10,88], NetPyNE[9] and PyNN[89]. The BSB aims to offer a broader set of tools that encompasses not just the high-level description of models like existing tools do, but also an architecture to accommodate well-designed reusable user code as components. In aggregate, the code-free descriptions of the components sum up to the model description. NeuroConstruct[90] fulfils a similar purpose to the BSB, but at the moment only supports extension in Java rather than Python.

On top of that, a subset of modeling problems remained unaddressed by existing frameworks, mostly related to the manipulation of neurons as individuals

(rather than populations) and as spatial entities: synthesis, elongation, pruning, or restricting morphologies to fit in the volume, individual determination of connection and synapse pairs, for example through intersection with other morphologies or special targets on those morphologies. The BSB addresses these needs with a set of user-friendly APIs designed to work with complex network topologies, cell morphologies and many other spatial and *n*-point problems. Intricate and heterogenous complexities of large-scale neural networks such as anisotropy, unique microstructures, or non-neuronal elements can be taken into account, so that any brain region may be modelled. These properties allow the BSB to fully empower a "scaffold" modeling strategy. The separation between model description and algorithm implementations makes scaffold models exceptionally easy to understand, and specific cell placement or connectivity datasets can be changed without having to regenerate the entire network.

There are 3 main phases in the scaffold modeling workflow that can be visited iteratively when changes need to be made: configuration, reconstruction and simulation. The core concepts of the framework during the reconstruction phase are i) the network *volume*, with the definition of various partitions such as layers, meshes or voxel sets (from brain atlases) and arranging elements which can be structured hierarchically to give rise a complex description of the entire region under consideration, ii) the *cell types* which determine the properties of cell populations, such as their spatial representation (soma radius, geometrical extension and/or morphologies) and density information, iii) the *placement* of said cell types into subspaces of the network volume using certain *placement strategies*, and iv) the *connectivity* between cell types using certain *connection strategies*. With this information, the framework places and connects the cells, storing the result in a network reconstruction file. Then the simulation phase follows, where *cell*

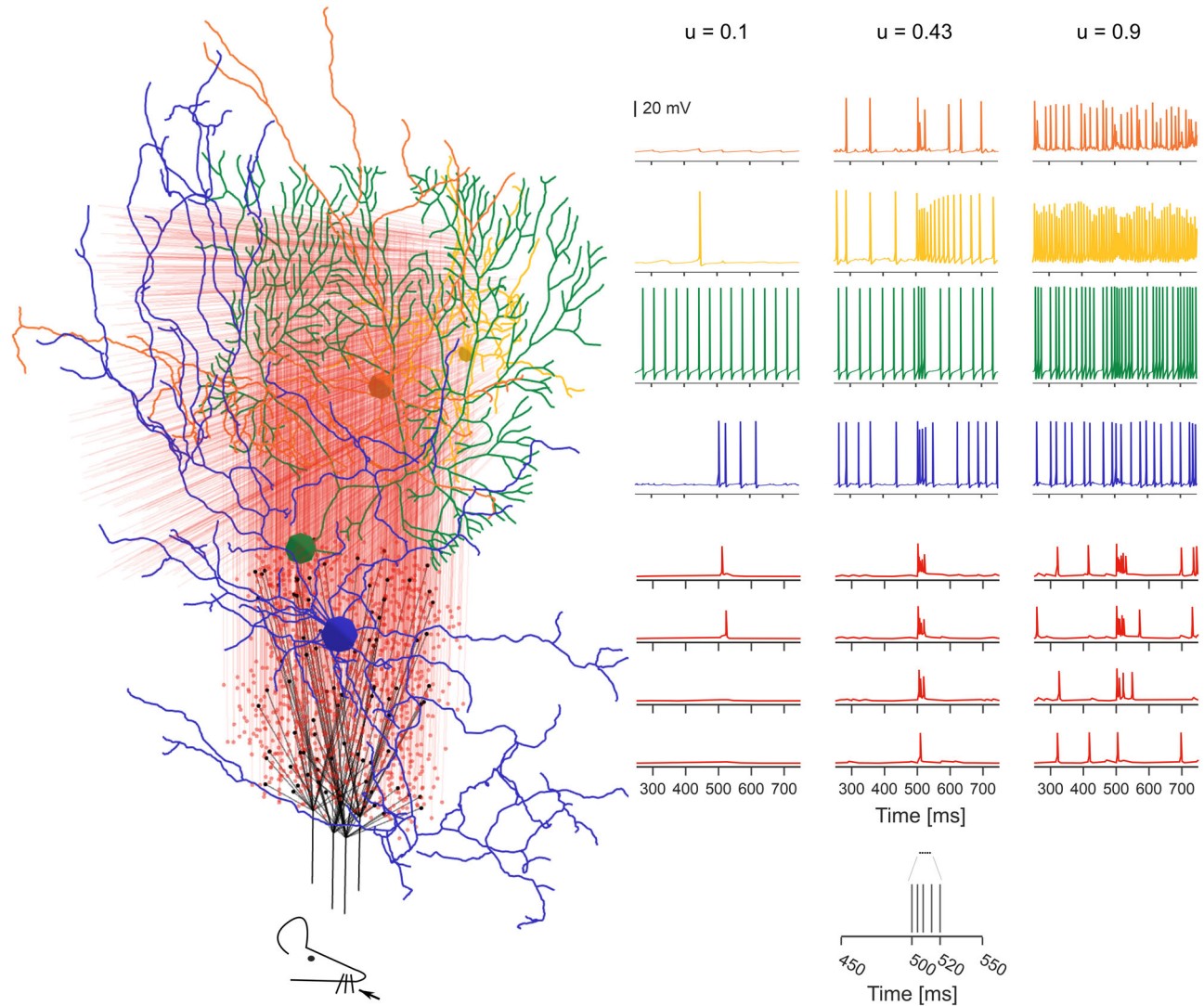

**Fig. 9 Activation of a vertical neuronal column in the cerebellar cortex.** A whisker air-puff stimulus (the *mf* burst) is delivered to 4 adjacent *mf*s, which branch in 4 *glom* clusters. *GrC*s respond rapidly with a burst when at least 2 dendrites are activated. A *GrC* dense cluster is formed and the signal propagates up through an *aa* bundle and transversally along a *pf* beam. *GoC*s receive the signal both on basolateral and apical dendrites. *PC*s vertically on top of the active cluster are invested by *aa* and *pf* synaptic inputs. On-beam *SC*s and *BC*s receive signals through *pf* synapses; *SC* axons inhibit mainly on-beam *PC*s, while *BC* axons inhibit mainly off-beam *PC*s. The membrane potential traces (*mf* burst starts at 500 ms) are shown for each neuronal population. Traces in the three columns correspond to three different release probabilities at the *mf-GrC* synapses: u = 0.1, u = 0.43 (control condition used in the rest of the paper), u = 0.9. The lower and higher u-values are typical of long-term synaptic depression and potentiation in the granular layer.

*models, synapse models* and *devices* define the simulator-specific representation of neuron and connection types and input/output variables (Fig. 1). The BSB is the only framework to offer arbitrary parallelization of non-parallel user codes through the interplay between network topology, a tiling paradigm (cutting the volume into smaller rhomboid pieces) determining the region of interest for each tile, and a process stitching the seams that the tiling might have created. All the steps of the scaffold workflow are available both as versatile CLI commands and library functions. The scaffold builder compiles its models into HDF5 or SONATA files, a format standard for neural models proposed by the Blue Brain Project and Allen Institute for Brain Science[91].

*Placement.* The placement is organized into placement objects that consider certain cell types, and a subspace of the volume. These objects determine the number, position, morphology and orientation of each cell, according to the desired placement strategy. A variety of configuration mechanisms exist to define the number of elements to be placed, such as a fixed count, a specific density (volumetric or planar) or a ratio to the density of another type. Other elements can be instantiated as well, with or without 3D positions for other purposes (e.g., fibers with their somatic origin outside the considered brain circuit). A post-processing step after placement may be enabled, where the elements can be pruned, moved, or labelled (e.g., labelling separate zones with their own connectivity patterns or identifying individuals to be hubs in a modular network). Each morphology can be rotated based on the voxel orientation in which it is placed, and fibers crossing multiple

voxels can be bent, in order to follow the surface folding of the region. The main placement strategies are (Fig. 1b):

*Particle placement.* The neurons are placed randomly and then checked for collisions, using kd-tree partitioning of the 3D space[92]. Colliding particles repel each other, the inertia of the particles is proportional to their radius. It is computationally efficient, yields uniform placement in 3D space, working properly even in irregular shapes, and it can deal with multiple cell types of different size. A pruning step can be enabled to remove cells positioned outside the desired subspace.

*Parallel array placement.* The neurons are placed in parallel rows on a desired surface, with a certain angle and specific distances between adjacent cells. A direction-specific jitter can be configured.

*Satellite placement.* The neurons are placed near each cell of an associated type (planet cells). Satellite positions are chosen at a random distance within a range based on the radii of the planet cells, so that each planet cell has a certain number of satellite cells around it.

*Connectivity.* Each connection identifies a presynaptic element and a postsynaptic element. When multi-compartmental neuron models are used, the synaptic locations on specific morphology compartments are also identified. Connections may

target either populations, subpopulations, or only specific regions of the cell morphologies. A post-processing step after connectivity may be enabled, where the identified synapse locations can be re-distributed (e.g. pruning or moving the synapses). The use of cell morphologies can be combined with soma-only approaches. Multiple synapses per pair can be requested, following a probability distribution.

The main connectivity strategies provided by the BSB implement different proximity-probability rules (Fig. 1c):

*Touch detection.* The 3D space is partitioned using a kd-tree to search for potentially intersecting cell pairs. Then, the actual points of intersection are determined using another kd-tree specific to the pair morphologies, with a maximum intersection distance parameter.

*Voxel intersection.* Each presynaptic cell is represented by a voxelized morphology and these voxels are tested for intersections with the voxels of the postsynaptic cells, using R-tree 3D space partitioning. When matching voxels are found, random compartments in each voxel are selected, introducing variability. It is, in this respect, less deterministic than the touch detection strategy[93]. A *Fiber intersection* variant exists to optimize the case of long, thin neurites, whose path can be deformed through space according to a 3D field of direction vectors.

*Distance-based in/out degree.* A probability distribution is applied to the distance between cells. Optionally 2 additional probability distributions can be given for the indegree and outdegree distribution of the network. The number of postsynaptic elements per presynaptic element is determined by samples of the outdegree distribution. Each postsynaptic target is weighted according to its distance probability, and the probability to transition from their current indegree N to indegree N + 1, as dictated by the indegree distribution. To optimize the algorithm, a kd-tree is queried for cells within a maximum search radius derived from the cumulative probability function of the distance distribution.

*Simulation.* The BSB can instruct simulators to run the configured models. Although multiple adapters to different simulators are provided (NEURON - RRID:SCR_005393; NEST - RRID:SCR_002963; Arbor - 10.5281/zenodo.4428108), there is no common high-level language to send instructions across simulators. Instead, sets of simulator-specific configuration expose the simulator underlying APIs more directly. These classes contain the simulator-specific logic to fully define inputs, execute, monitor progress, and collect output of simulations. The interface to the NEURON simulator has been applied specifically in this work.

NEURON[94] cooperates with our Python packages: *Arborize* to create high-level descriptions of cell models [https://github.com/dbbs-lab/arborize], *Patch* to provide a convenience layer on top of NEURON [https://github.com/Helveg/patch], and *Glia* to manage NMODL file dependencies and versioning [https://github.com/dbbs-lab/glia]. Together, these packages and the NEURON adapter provide out-of-the-box load balanced parallel simulations in NEURON. The adapter is capable of creating and connecting these *arborized* cell models over multiple cores, implements device models such as spike generators, voltage and synapse recorders and collects the recorded measurements in an HDF5 result file. The recorders can specify targets at the cellular or subcellular level, recording membrane or synapse voltages, conductances, currents and ionic concentrations. These easy configurable devices allow to monitor all signals propagating across the network to reproduce results at multiple scales.

*Visualization.* The BSB provides a plotting module to directly visualize simulation results including 3D network plots, cell activity in 3D space, PSTH, raster plots, synaptic currents mapped on cell morphologies, and more. The BSB provides a Blender module containing a complete blender pipeline for rendering videos of the network activity on a single machine or a cluster. The BSB can be used in Blender Python environment and provides functions to synchronize the state of the network with the Blender scene, to animate results or to generate *debug frames*, to troubleshoot placement, connectivity or simulation issues.

**The cerebellar cortical model.** Using the BSB, a mouse cerebellar cortical microcircuit was reconstructed and simulated. The example reported here refers to a volume partitioned in a granular, Purkinje and molecular layer. Specifically, the volume extended 300 μm along *x*, 200 μm along *z*, and 295 μm along *y* (*y* = layer thickness; 130 μm granular layer, 15 μm Purkinje cell layer, 150 μm molecular layer). In the reference system, *x-y* is the sagittal plane, *x-z* the horizontal plane, *z-y* the coronal plane. The reconstructed volume was 17.7·10⁻³ mm³. The model was filled with biophysically detailed compartmental neurons for each cell type. Some structural data and multiple observations from electrical recordings in vivo and in vitro were used as constraints in building the model, further experimental measurements were used for structural and functional validation.

*Neuron placement.* Both the granular layer and the molecular layer were filled using *particle placement*. The granular layer is made up of densely packed granule cells (*GrC*) and glomeruli (*glom*) intercalated with Golgi cells (*GoC*). Furthermore, a certain number of mossy fibers (*mf*) was created (without any 3D position), each

terminating in about 20 glomeruli. Each *GrC* emits an ascending axon (*aa*) that raises perpendicularly to the overlying cerebellar surface and reaches the molecular layer bifurcating into two opposite branches of a parallel fiber (*pf*) elongating on the *z*-axis (major lamellar axis).

The molecular layer was divided into a superficial sublayer (2/3 of the thickness) hosting the stellate cells (*SC*) and a deep sublayer (1/3 of the thickness) hosting the basket cells (*BC*)[24,95].

The Purkinje cells (*PC*) were placed on a horizontal plane (*x-z*) using *parallel array placement*. *PC*s were placed along parallel lines, with an inclination angle of about 70° with respect to the major lamellar axis. The dendritic tree of the *PC* is flattened on the sagittal plane and extends for about 150 μm[20]. The parallel arrays were placed at such a distance that the *PC* dendritic trees did not overlap, while along the major lamellar axis their somata could be packed closely together. For each neuronal population, the nearest neighbour, the pairwise distance distribution, and the radial distribution function were computed.

*Knowledge base for microcircuit connectivity.* This chapter summarizes the fundamental knowledge used to reconstruct cerebellar microcircuit connectivity and highlights which parameters are reported or absent in literature, implying the different strategies adopted in the BSB.

The connectivity of *mf*s and *glom*s is supported by an extended anatomo-physiological analysis indicating that (i) each *mf* spreads the input signal into a cluster of *glom*s[25], (ii) each *GrC* sends its 4 dendrites to *glom*s belonging to different *mf*s within about 30 μm[26], (iii) 1 excitatory synapse is formed on the terminal compartment of each *GrC* dendrite[27,28], (iv) 1 excitatory synapse is formed on *GoC* basolateral dendrites[31], (v) 1 inhibitory synapse is formed on the preterminal compartment of each *GrC* dendrite[96].

The connectivity of *GoC*s is also supported by a robust experimental dataset. (i) *GoC*s receive an undetermined number of excitatory inputs from *mf*s through *glom*s, that in turn host ~2 *GoC* basolateral dendrites each[31]. (ii) Given that *GoC-GrC* synapses are inside *glom*s, each *GrC* dendrite receives inhibition from a *GoC* whose axon reaches the *glom* contacting that dendrite[29,30]. (iii) *GoC*s may receive as many as ~400 *aa* synapses on basolateral dendrites and ~1200 *pf* synapses on apical dendrites[32]. (iv) *GoC*s make GABAergic synapses onto other *GoC*s[33], but their number was not reported. (v) There are 2-4 gap junctions per *GoC* pair[34]. Therefore, geometrical rules were used to extract the missing parameters.

The connectivity of *PC*s can be derived by the axons intercepting their dendritic tree. (i) There is one single synapse per *pf-PC* crossing. As a whole, the number of *pf-PC* synapses may range up to ~100′000 [35], many of which would be silent[97]. Based on spines density[98] and the total length of a *PC* dendritic tree[18,20], the number of possible *pf* synapses was estimated to be 15′000-20′000, with a minimum of ~100 synapses needed to generate a simple spike[36]. (ii) The number of *aa-PC* synapses is not known but there would be multiple synapses per *aa-PC* pair[36].

The connectivity of *MLI* is not completely defined. It is known that (i) *MLI*s receive excitatory input from *pf*s, (ii) *MLI*s form inhibitory connections with other *MLI*s of the same type[37], (iii) collaterals of a *SC* axon mainly extend on the coronal plane, while collaterals of a *BC* axon mainly extend on the horizontal plane, innervating multiple *PC*s[38], (iv) each *PC* receives 3-50 baskets around the soma from as many different *BC*s[39], while *SC* axons terminate on intermediate *PC* dendritic branches with 0.3-1.6 μm diameter[38,62,66].

*Selection of microcircuit connectivity rules.* The connectome of the cerebellar network took into consideration 16 connection types (identified by their source and target neuronal population): *mf-glom, glom-GrC, glom-GoC, GoC-GrC, GoC-GoC, GrC(aa)-GoC, GrC(aa)-PC, GrC(pf)-GoC, GrC(pf)-PC, GrC(pf)-SC, GrC(pf)-BC, SC-PC, BC-PC, SC-SC, BC-BC, GoC-GoC, GoC-GoC gap* junctions.

Glomerular connectivity is a special case since it is largely constrained by prescribed neuroanatomical and neurophysiological information. Then, since the *glom* did not have a defined morphological model, they were connected through probability strategies (*distance-based in/out degree*) to identify nearby compartments for synaptic locations on the target cell types, for which a realistic morphology was used.

Specifically, glomerular connectivity was largely based on prescribed anatomo-physiological rules:

*mf-glom.* The *mf* arborization creates anisotropic clusters of glomeruli and clusters originating from different *mf*s mixed up with each other to some degree[25]. Taking into account short branches since the small reconstructed volume, a local branching algorithm grouped glomeruli (20 ± 3 per cluster, normally distributed) receiving signals from the same *mf* by a distance-based probability rule.

*glom-GrC.* For each *GrC*, a pool of nearby *glom*s were selected based on the distance between the *glom* barycentre and the *GrC* soma center, and a maximum extension of *GrC* dendrites of 30-40 μm[28]. From the pool, 4 *glom*s, each from a different cluster, were randomly sampled and connected with one of the 4 *GrC* dendrites.

*glom-GoC.* For each *glom*, all *GoC*s with their soma at a radial distance less than 50 μm (corresponding to an average extension of *GoC* basolateral dendrites, isotropically in 3D[14,32]) were connected. The synapse was placed on a

basolateral dendrite using an exponential distribution favouring the compartments closer to the centre of the *glom*.

*GoC-GrC*. This connection absorbed the connection *GoC-glom* which was generated using 3D proximity and a mean divergence (out-degree) of 40; then, each *GoC* synapsed directly on all *GrC*s that shared those *glom*s.

The rest of the cerebellar connectome was reconstructed by applying *voxel* and *fiber intersection*, and *touch detection*. *Voxel intersection* was preferred when 3D morphologies were intersecting. This strategy, by introducing a cubic convolution and randomization, reduced overfitting artifacts arising from the intersection of identical morphologies arranged in the quasi-crystalline cerebellar microcircuit. *Touch detection* and *fiber intersection* were preferred when dealing with 2D fibers (i.e. *aa* and *pf*), for which the *voxel intersection* would create cubic volumes not representative of the fibers as line segments. Therefore, for the connections involving *aa*s (*GrC(aa)-GoC* and *GrC(aa)-PC*), *touch detection* was applied with a tolerance distance of 3 μm. For the connections involving *pf*s (*GrC(pf)-GoC*, *GrC(pf)-PC*, *GrC(pf)-SC*, *GrC(pf)-BC*), *fiber intersection* was applied with affinity =0.1 and resolution =20 μm. For the other connections, *voxel intersection* was applied: *GoC-GoC*, *SC-SC*, and *BC-BC* with affinity =0.5; *SC-PC* with affinity =0.1; *BC-PC* with affinity =1, *GoC-GoC* gap junctions with affinity= 0.2. For chemical synapses, the presynaptic compartment was always axonal and the postsynaptic compartment was dendritic or somatic (as for the *BC-PC* connection). For electrical synapses, gap junctions were created between dendrites. Following biological indications, specific sectors of morphologies were selected as source or target for synaptic localization, and, for each connected cell pair, the desired number of synapses was defined, eventually as a normal distribution (mean ± sd) (see table in Fig. 2d).

Specific sectors of the dendritic trees of *GoC* and *PC* were used as targets for synapse formation. *GoC*s receive inhibitory and electrical synapses from other *GoC*s as well as *aa* synapses on basolateral dendrites in the granular layer, while *pf* synapses impinge on apical dendrites in the molecular layer[32]. *PC*s receive *aa* and *pf* synaptic inputs on different parts of the dendritic tree, and also *SC*s and *BC*s target different parts of the neuron[62,66]. (i) *GrC(aa)* ➔ *PC* dendrites with diameter < 0.75 μm[18,20,40]. (ii) *GrC(pf)* ➔ *PC* dendrites with diameter between 0.75 and 1.6 μm. (iii) *SC* ➔ *PC* dendrites with diameter between 0.3 and 1.6 μm[38,62,66]. (iv) *BC* ➔ *PC* soma.

*Multi-compartmental neuron models and synaptic models*. Detailed multi-compartmental models of *GrC*, *GoC*, *PC*, *SC*, *BC* are available, in which dendritic and axonal processes are endowed with voltage-dependent ionic channels and synaptic receptors (Table S2). In each model, cell-specific aspects critical for function are reproduced, e.g., the role of the axon initial segment, spontaneous firing and burst-pause behaviour. The following receptor-channel models, all validated against in vitro recordings, and generic gap junction models were inserted in the appropriate neuron compartments:

*GrC* synapses[17,99]. *mf-GrC*: AMPA and NMDA receptors; *GoC-GrC*: GABAalpha1/6 receptors.
*GoC* synapses[16]. *pf-GoC*: AMPA; *aa-GoC*: AMPA and NMDA; *mf -GoC*: AMPA and NMDA; *GoC-GoC*: GABAalpha1, gap junctions[34,100].
*PC* (Z + type) synapses[62]. *pf-PC* and *aa-PC*: AMPA; *SC-PC* and *BC-PC*: GABAalpha1.
*SC* and *BC* synapses[19]. *pf -SC* and *pf -BC*: AMPA and NMDA; *SC-SC* and *BC-BC*: GABAalpha1.

Chemical neurotransmission was modelled using the Tzodyks and Markram scheme[101,102] for neurotransmitter release, and receptors kinetic schemes for postsynaptic receptor activation. Glutamatergic neurotransmission could activate either only AMPA or both AMPA and NMDA receptors[103]. GABAergic neurotransmission activated GABA-A receptors[104]. A neurotransmitter impulse was followed by a slow diffusion wave generating both a transient and a sustained component of the postsynaptic response, as observed experimentally. Parameters describing release probability, diffusion, ionic receptor mechanisms, vesicle cycling, recovery time constant, electrical conduction were derived from the corresponding original papers.

*Network simulations: stimulation and analysis*. All simulations used the NEURON adapter of the BSB and were run in parallel through MPI on the CSCS *Piz Daint* supercomputer, with a time resolution of 0.025 ms. Simulations started with a 5-s stabilization period followed by a 100 ms initialization period, in which random *mf* inputs desynchronized the network. In all simulations, spikes and voltage traces at soma of all neurons were recorded. Depending on specific analyses, in some simulations further microscopic variables were recorded, as explained below. A set of stimulation protocols reproducing specific spatio-temporal patterns of *mf* activity was used to functionally validate the cerebellar network model; in some cases, the protocols were repeated using an altered version of the network model in terms of connectome ("by-lesion" approach), to quantitatively check the relative roles of the connection types.

Diffused background stimulation: The cerebellum in vivo is constantly bombarded by a diffused background noise, which determines the resting state activity of neurons and is thought to entrain the network into coherent low-frequency oscillations[47,58,59]. Therefore, we first explored the response of the network model

to a random Poisson noise at 4 Hz[22] on all *mf*s for 4 seconds, proving a testbench to validate the structural and functional network balance.

*Steady state analysis*. We compared basal discharges in the network to those recorded at rest in vivo. The mean frequency of each population was computed. *Oscillatory state analysis*. We investigated the emergence of low-frequency coherent oscillations in the *GoC* and *GrC* populations. The power spectrum of *GoC* and *GrC* firing activity was computed by Fast Fourier Transform (FFT), applied to time-binned spike-counts (2.5 ms bins). The zero-component was cut off and the FFT was smoothed using a Savitzki-Golay filter (6th order polynomial, window of 51 bins). The same analysis was performed when *GoC-GoC* gap junctions were disabled, in order to check the role of electrical coupling in oscillatory behaviour of the granular layer.

*Coupling and synchrony*. This analysis involved the calculation of the coupling degree and of spike correlations and was applied to *GoC*s. *GoC*s can have either direct coupling, when two cells communicate directly via gap-junctions or indirect coupling otherwise. The coupling degree was evaluated as the electrotonic distance in cell pairs[49], which was calculated as the inverse of the number of gap-junctions on the shortest path between any *GoC* pairs in the network. According to this metric, a higher distance means to have less gap junctions or more *GoC*s in between. The cross-correlogram was calculated with a 0.5 ms time bin considering all pairs located <100 μm distance from each other, i) on pairs with direct coupling, ii) on pairs with indirect coupling, and iii) all pairs when gap junctions were disabled. The relation between the electrotonic distance and the maximum cross-correlation of spikes in the pairs was calculated in different conditions that include: (i) 4-Hz Poisson mossy fibre activity (control), (ii) disabling the gap junctions, (iii) random spike times (when the same number of spikes per *GoC* were assigned uniformly random times in the same interval, to serve as a non-synchronous baseline). The percentage of synchronous spikes was calculated using different time-lag windows and the probability density of spike coincidence (with ± 5 ms windows) was spatially mapped on the granular layer plane.

mf burst stimulation: The cerebellum in vivo responds with localized burst-burst patterns to facial or whisker sensory stimulation[22,54]. These bursts are supposed to run on collimated *mf* bundles generating dense response clusters in the granular layer and thereby activating the neuronal network downstream[21,23,52,99]. To simulate this functional response, we delivered a *mf* stimulus burst, superimposed on background noise at 4 Hz, to 4 *mf*s in the center of the horizontal plane, activating about 80 *glom*s. The *mf* burst lasted 20 ms and was made of 5 spikes at fixed time instants (on average 200 Hz, maximum 250 Hz), within range of in vivo patterns[22,40,54]. Ten simulations were run to account for random variability of the background input and the network responses. Multiple variables over time were recorded: spike times and membrane voltages of every cell, synaptic currents in the dendrites of some cells, the internal calcium concentration $[Ca^{2+}]_{in}$ in the dendrites of all *GrC*s and of some *GoC*s. Further simulations were carried out using different values of neurotransmitter release probability at the *mf-GrC* synapse (from u = 0.43 to u = 0.1 and to u = 0.9).

*General analysis of response patterns*. For each neuronal population, a raster plot and a PSTH (peri-stimulus time histogram) was computed. Each population was described using Multiple Regression Analysis: the dependent variable was the average firing frequency during 40 ms after the stimulus onset, over 10 simulations, and the independent variables were the average numbers of spikes received from each presynaptic population. The linear regression was reported as direction coefficients and $R^2$ score.
*Analysis of granular layer responses*. For *GrC*s we related the number of dendrites activated by the *mf* burst with the number of output spikes and the first spike latency. The same protocol was carried out while switching-off phasic and tonic inhibition from *GoC*s (GABA-A receptor blockade). This allowed us to investigate excitatory-inhibitory loops in the granular layer, by estimating the response patterns of *GrC*s, their latencies, and the fraction of *GrC*s activated compared with the control condition. Furthermore, for each *GrC*, the level of $[Ca^{2+}]_{in}$ in the dendrites averaged on 500 ms from the *mf* burst was extracted and these $[Ca^{2+}]_{in}$ values were related to the number of dendrites activated by the stimulus. The correlation analysis used Normalized Mutual Information (NMI)[105].
*Analysis of PC responses*. The *PC* response was analyzed to evaluate the burst-pause behavior. For each *PC*, an automatic algorithm extracted any shortening of the inter-spike intervals during the stimulus window (*burst coeff.*) and any elongation after the stimulus (*pause coeff.*) compared to baseline. The *burst coeff.* was correlated with the number of excitatory synaptic inputs (from *pf*s and *aa*s) by multiple regression analysis. The *pause coeff.* was correlated with the number of inhibitory synaptic inputs (from *MLI*s) received during the burst stimulation (20 ms *mf* burst + 20 ms of delayed effects), by calculating the Normalized Mutual Information (NMI). Furthermore, the relation between the burst and pause coefficients themselves was analysed, by NMI. Further simulations were run clamping an on-beam *PC* at −70 mV, recording all synaptic currents. All excitatory synaptic currents (AMPA from *aa*s and *pf*s) and all inhibitory synaptic currents (GABA from *SC*s and *BC*s) were summed

up, then the cross-correlation among these two rectified and normalized currents was calculated, to identify the time lag between them.

*Visualization of subcellular variables.* In some cases, ad-hoc computationally expensive recordings of multiple microscopic variables were performed. In an example focused on a *GoC*, all synaptic currents (AMPA and NMDA from *gloms* on basolateral dendrites, AMPA and NMDA from *aas* on basolateral dendrites, AMPA from *pf*s on apical dendrites, GABA from other *GoC*s on basolateral dendrites), and $[Ca^{2+}]_{in}$ were recorded and animated (see *Visualization*).

Lateral Poisson stimulation: The lateral inhibition from *MLI*s to *PC*s comes from activated *MLI*s providing inhibition to off-beam *PC*s, mainly from *BC*s due to their axon orientation[44]. To simulate this functional response, we delivered a 50 Hz Poisson distributed stimulus, lasting 100 ms, superimposed on the background noise (at 4 Hz), on 24 *mf*s on one side of the volume, to monitor the modulation of *MLI* inhibitory effects on *PC*s at different distances from the active cluster. Two conditions were evaluated: i) control and ii) *MLI*s disconnected from *PC*s. Ten simulations for each condition were carried out.

*Analysis of PC responses.* For each *PC*, the average Inter-Spike-Interval (ISI) during 200 ms baseline and the average ISI during the 100-ms stimulus was computed. The relationship between the distance of a *PC* from the active cluster and its activity modulation (balance between *GrC*s excitation and *MLI*s inhibition) was investigated in control condition and in the "no *MLI-PC*" condition.

Disabled MLI inhibition on PCs: Feedforward inhibition from molecular layer interneurons regulates adaptation of the vestibulo-ocular reflex, as shown in behaving mice with GABA$_A$ receptor–mediated synaptic inhibition selectively blocked from Purkinje cells[64]. To simulate this condition and functional response, we disabled the inhibitory synapses on PCs and delivered a single impulse to a bundle of 13 *mf*s. The temporal dispersion (jitter) of evoked Purkinje cell simple spikes was quantified as the standard deviation of spike latency in a 10-ms window following stimulation.

Long-term plasticity at pf-PC synapses: Simple spike suppression of PCs is the main outcome of long-term plasticity mechanisms involved in learning tasks, as eye blink classical conditioning paradigm[65]. To test the effect of LTD, we delivered an input of 50 Hz[106] lasting 160 ms on a bundle of 4 *mf*s and set different values of synaptic conductance ($g_{syn}$) at AMPA-mediated pf-PC synapses. The default value of 1200 pS was lowered progressively to 30% in 10 steps (5 simulations were performed for each case and the average of results was taken). The PC firing modulation was computed as the difference between the frequency during stimulus and the baseline frequency for each PC, in pre-learning ($g_{syn}$ default value) and in post-learning (after LTD). The amount of suppression was then estimated as ratio between firing modulation in post-learning and the one in pre-learning.

## Statistics and reproducibility

*Analysis of population responses and oscillations to sensory burst stimulation.* A 40 ms window, beginning at the onset of the stimulus, of spikes at the soma of each cell was considered and combined with the connectome, to determine how many spikes each cell received during the window. Across 10 simulations, the mean amount of spikes per cell was aggregated. Linear regression provided by `sklearn.linear_model.LinearRegression` was applied with the mean ($n = 10$ simulations) number of spikes produced during the window per cell as independent variable, and the mean ($n =$ the same 10 simulations) amount of spikes received as a dependent variable per presynaptic cell type. E.g.: The Golgi cell receives excitatory glomerulus and Granule cell stimulation, and inhibiting Golgi cell stimulation. So the mean amounts of glomerulus spikes, granule cell spikes, and Golgi cell spikes, were compared to the amount of spikes each Golgi cell produced. This allows a factor and correlation coefficient to be established between each cell type: The factor tells us how many presynaptic spikes are required per additional or suppressed spike of the postsynaptic cell, and the correlation coefficient tells us the confidence we can place in the observation in the data.

The oscillations were analysed by time-binning the population spikes in 2.5 ms bins, and applying a discrete Fast Fourier Transform provided by `scipy.fft`, using Blackmann windowing (parameter-free) to reduce spectral leakage, and Savitzky-Golay filtering (window width 51 bins, 6th order polynomial fitting) to filter and smooth the results to increase the signal to noise ratio, all provided by `scipy`.

*Golgi cell spike coincidence analysis.* Coincidence matrices were constructed by creating a square matrix representing all Golgi-to-Golgi pairs, and comparing each pair's spikes produced at the soma. All spike pairs less than dt time separated from each other counted towards the coincidence value in that cell pair's M(dt) coincidence matrix value. This value is then divided by the amount of spikes in the first cell. So a spike train [0, 1, 2, 5.8] compared to a spike train [1.1, 1.2, 4, 6, 7] would yield a coincidence of 3, for dt = 0.3.

The knockout conditions were always obtained by removing the gap junctions, and the random conditions were always obtained by assigning a new uniformly

random timestamp (within the same time interval) to each spike in the dataset, and repeating identical procedures.

The Z-scores (Fig. 4a) were calculated by binning the spikes into 5 ms bins, and counting the amount of bins that contained spikes in both cells. Now offsetting the signals from −100ms to 100 ms in 0.5 ms steps, obtaining 400 different overlapping bin values, the Z-scores were calculated for these 400 values. The maximum z-score, the best overlap, was retained per cell pair. A sliding-window average of the electrotonic distance (width = 0.2) versus the max-z score is shown. Figure 4b shows the average Z-score for all selected pairs with a time lag from −5 to 5 ms in 100 steps. The "direct pairs" were selected by taking all Golgi cell pairs that form synapses with another, the "indirect pairs" by selecting all other pairs.

The spike coincidence line plot (Fig. 4c) was constructed by, per time lag window on the X-axis, summing up all the coincidence matrix values, and dividing them by the number of spikes produced in the population, yielding a coincident spike pairs / total spikes ratio. The error bars show the standard error of the mean for $N = 1181$ pairs.

To investigate the spatial relationship of spike coincidence, a kernel density estimation of the coincidence matrix was performed by taking each Golgi-Golgi pair's relative position to each other (XY position cell 1 minus XY position cell 2) (Fig. 4d). During KDE, each pair's relative position was then weighted by the coincidence matrix value, normalized by the amount of spikes produced by cell 1. The KDE was provided by `scipy.stats.gaussian_kde`. Contour plotting was provided by `plotly.Contour`

*Burst pause analysis.* The burst pause plots (Fig. 5b, all 3 panels) were created by following the protocol described in the Methods to obtain the burst and pause coefficients. The coefficients were then used as the dependent variable with `sklearn.linear_model.LinearRegression` providing the linear regression.*Feedforward inhibition analysis*

The synaptic currents were aggregated by taking all the synapses on a cell of the same type (AMPA, GABA) and summing them up to the total current for 1 simulation. The mean ($n = 10$ simulations) was calculated. The averages were normalized (divided by the max value for the positive GABA current, and by the min value for negative AMPA current). Cross correlation of the 2 signals was performed provided by `numpy.correlate`.

*Lateral inhibition analysis.* The GrC activation was calculated by kernel density estimation of the all the granule cell spikes, with each spike becoming an equally weighted sample in the KDE on the XY position where the spike occurred. The estimated density was then calculated for each granule cell based on their XY position, and all granule cells binned into 30 equal bins (10um bins) according to their X position. The GrC activation is the normalized mean activation per bin.

The relative change in Purkinje cell activity was calculated as stimulus activity divided by baseline activity minus one, to show the relative increase in activity during stimulus. The baseline is measured by taking the mean ($n = 10$ simulations) frequency of spikes during a 200 ms window in absence of any stimulation of the network per cell, and the stimulus activity as the mean ($n = 10$ simulations) frequency during a 100 ms window starting from stimulus onset. This was plotted versus the X position of each Purkinje cell, and Kernel Ridge Regression (KRR, provided by `sklearn.kernel_ridge.KernelRidge`, with RBF kernel. Alpha and gamma parameters were optimized with a grid search by `sklearn.model_selection.GridSearchCV`, alpha between 0.001 and 1 in 4 steps, and gamma between 10e-100 and 10e20 in 200 logarithmically spaced steps).

*Feedforward jitter analysis.* To find the spread of the Purkinje cell population ($n = 91$) response to a single stimulus, a probability density function (PDF) of their first spike response to the stimulus was calculated. The spikes were selected by taking the population spikes of the Purkinje cells, binning them and selecting the spikes belonging only to the largest possible interval around the maximum bin that would not contain any empty bins. A normal distribution was fitted on all the spikes belonging to the selected bins, fitting provided by `scipy.stats.norm.fit`, and then extrapolated to the entire visualised interval.

*Purkinje plasticity modulation analysis.* To measure PC activity, for the stimulus-response a 150 ms window starting at the stimulus was considered, and a 300 ms window before the stimulus used as baseline comparison. The mean ($n = 14$ cells) absolute increase in firing frequency (stimulus-response frequency minus baseline frequency) and the standard error of the mean were reported for each plasticity condition.

**Reporting summary**. Further information on research design is available in the Nature Portfolio Reporting Summary linked to this article.

## Code availability

The BSB can be cited from https://doi.org/10.5281/zenodo.7243999. The source code is available at https://github.com/dbbs-lab/bsb.

The code of the analysis of the data, and generation of the graphs is available at https://github.com/Helveg/deschepper-etal-2022-source, including the steps needed to reproduce the findings (README).

The exact versions of each used package is included in the text file "requirements.txt". The BSB directly depends on NumPy[107], SciPy[108], Scikit-learn[109], rtree[110], pynrrd, MorphIO[111] and mpi4py[112].

## Data availability

All datasets obtained from reconstruction and simulations are available at the following DOIs:
Morphology repository and reconstructed network
- https://doi.org/10.5281/zenodo.7230455
- https://doi.org/10.5281/zenodo.7230288
Simulation protocols and results
- https://doi.org/10.5281/zenodo.7235526
- https://doi.org/10.5281/zenodo.7235462
- https://doi.org/10.5281/zenodo.7235428
- https://doi.org/10.5281/zenodo.7235051
- https://doi.org/10.5281/zenodo.7235095
- https://doi.org/10.5281/zenodo.7230798
- https://doi.org/10.5281/zenodo.7230830
- https://doi.org/10.5281/zenodo.7230836
- https://doi.org/10.5281/zenodo.7231068
- https://doi.org/10.5281/zenodo.7231161
- https://doi.org/10.5281/zenodo.7231187
- https://doi.org/10.5281/zenodo.7230239
- https://doi.org/10.5281/zenodo.7248336

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

This research has received funding from the European Union's Horizon 2020 Framework Program for Research and Innovation under the Specific Grant Agreement No. 945539 (Human Brain Project SGA3) and Specific Grant Agreement No. 785907 (Human Brain Project SGA2) and from Centro Fermi project "Local Neuronal Microcircuits" to ED. We acknowledge the use of EBRAINS platform and Fenix Infrastructure resources, which are partially funded from the European Union's Horizon 2020 research and innovation programme through the ICEI project under the grant agreement No. 800858. Several open source contributions to the project were made, summarized on https://github.com/dbbs-lab/bsb/graphs/contributors.

## Author contributions

C.C. and R.D.S. designed and developed the informatic framework and performed the simulations; R.D.S. wrote most of the code; A.G., S.M., M.R., A.A. contributed with essential model components; R.D.S., C.C., E.D. analyzed the data, wrote the manuscript, and prepared the figures; C.C. and E.D. coordinated the work and the EBRAINS interaction. E.D. promoted the project, supported it financially, defined the physiological aspects and finalized the manuscript.

## Competing interests

The authors declare no competing interests.

## Additional information

Acknowledgements

