## [Peer Review File · Communications Biology]

Reviewers' comments:

Reviewer #1 (Remarks to the Author):

The manuscript "Model simulations unveil the structure-function-dynamics relationship of the cerebellar cortical microcircuit" by De Schepper and colleagues describes a software tool for flexibly creating and modifying cerebellar cortical multicompartmental cell networks. The specific simulator, Brain Scaffold Builder (BSB), allows the user to build cerebellar cortical-specific microcircuits. The authors then demonstrate the capabilities of their tool by building a subnetwork of the cerebellar cortex with ~30000 neuronal elements and 1.5 million synapses, and perform simulations to monitor the in silico network activity with subcellular precision under in vivo-like conditions.

The software is Python based, which is in line with current standards of computational neuroscience, and runs simulations in NEURON, the most commonly used multicompartmental neuron simulator. The subnetwork computations of the cerebellar cortex have been intensely studied in the past decades and with the recent improvements of mesoscale in vivo recording techniques, the modeling of the recorded neuronal ensembles are important to either validate experimentally unmeasurable features of the network or to make future predictions about planned experiments' outcome. Therefore, the presented work is of great importance, however, I have two major and two minor comments, which are the following.

Major

1) There is pre-existing work of this kind (i.e. constructing networks with realistic 3D cell morphologies and connectivity patterns). The neuroConstruct (Gleeson et al., 2007, Neuron; see also at neuroconstruct.org) uses neuroML (neuroml.org) to define cell types, synapses, connectivity motifs and plasticity mechanisms, which is a platform independent model description language, therefore it can be used with other simulation platforms as well. A similar, Python-interfaced tool is Moose (<https://moose.ncbs.res.in/readthedocs/index.html>), which can operate at multiple levels of detail ranging from biochemical reactions in subcellular components to large networks of neurons. How does BSB compare (or complement) these existing software tools? What are the new features introduced by BSB?

2) The authors simulated two scenarios of the cerebellar subnetwork constructed by BSB. The first is the resting state activity and the second is the impulse response of the cerebellar network. The results of their model simulations are in good agreement with previously published in vitro or in vivo physiological recordings. I'm curious whether their model could also reproduce finer scale experimental findings, such as the one by van Welie and colleagues (2016, Neuron), where they show millisecond synchrony between Golgi cells in vivo, which requires gap junctions and is enhanced by sensory stimuli.

Minor

1) It would be helpful to have a comprehensive supplementary table of the passive electrical properties and the active conductances of the cell types in the network model to see how these parameters compare to previously published data to achieve the reported physiological resting state activity of the cerebellar network.

2) There is a typo on page 4, line 124 after the word "volume" in the references.

Reviewer #2 (Remarks to the Author):

The authors have developed a methodological framework utilizing the Brain Scaffold Builder to construct a 'realistic' microcircuit model of the cerebellum with all known cell components, morphologies, synaptic properties, connectivities and physiological responses. Their model contains a little over 30,000 neurons, a very small subset of the cerebellar circuit. They claim that the scope of their study is unveil the structure-function-dynamics of the cerebellar microcircuit.

I am afraid they have not succeeded in their goal:

1) To accurately simulate the structure of a cerebellar microcircuit they utilized experimental data from various sources. The authors emphasized that by doing so the model's realism is increased. Such experimentally recorded data are full of errors, which if are not appropriately handled, but instead are used as such to build a computational model, then they are inserted into the model too. So, what techniques have the authors used to wrangle with the experimental structure data?

2) I failed to see any functional predictions the model made. We know that the cerebellum plays functional roles in cognition and motor control. Can the simulated cerebellar microcircuit simulate any of these functionalities? If so, how does the microcircuit model computes information that support any of these functionalities? We also know that the cerebellum supports learning (LTD) via the parallel fibers and Purkinjee cell dendrites (one pathway). There are many other types of learning observed in the cerebellum. How do these forms of learning provide insightful discoveries to the cerebellum's supported functions?

3) To accurately simulate the neuronal dynamics of the various cellular components of the microcircuit I would have preferred to see side-by-side comparisons of the model's outputs with experimental data under preferably the same experimental conditions. What optimization algorithm is used to fit the model's cellular responses to the experimentally recorded ones? Otherwise, if the group has not access to recorded traces of cellular responses, then perhaps they can calculate features (spike amplitude, duration, ISI, membrane tau, input resistance, etc) of the simulated cellular responses and compare them to same feature values of experimentally derived responses. This way further biological realism may be accomplished.

4) It is unclear where previous modelling attempts of the cerebellum failed, and how this model succeeded in overcoming these failures. Perhaps you may include such a section in your Discussion section.

Reviewer 1

The manuscript “Model simulations unveil the structure-function-dynamics relationship of the cerebellar cortical microcircuit” by De Schepper and colleagues describes a software tool for flexibly creating and modifying cerebellar cortical multicompartmental cell networks. The specific simulator, Brain Scaffold Builder (BSB), allows the user to build cerebellar cortical-specific microcircuits. The authors then demonstrate the capabilities of their tool by building a subnetwork of the cerebellar cortex with ~30000 neuronal elements and 1.5 million synapses, and perform simulations to monitor the in silico network activity with subcellular precision under in vivo-like conditions.

The software is Python based, which is in line with current standards of computational neuroscience, and runs simulations in NEURON, the most commonly used multicompartmental neuron simulator. The subnetwork computations of the cerebellar cortex have been intensely studied in the past decades and with the recent improvements of mesoscale in vivo recording techniques, the modeling of the recorded neuronal ensembles are important to either validate experimentally unmeasurable features of the network or to make future predictions about planned experiments' outcome. Therefore, the presented work is of great importance, however, I have two major and two minor comments, which are the following.

We thank the reviewer for the very useful comments that allowed us to substantially improve the manuscript. The new and the significantly revised parts are highlighted in blue.

Reviewer 1 comment 1 - Comparison to other tools

There is pre-existing work of this kind (i.e. constructing networks with realistic 3D cell morphologies and connectivity patterns). The neuroConstruct (Gleeson et al., 2007, Neuron; see also at neuroconstruct.org) uses neuroML (neuroml.org) to define cell types, synapses, connectivity motifs and plasticity mechanisms, which is a platform independent model description language, therefore it can be used with other simulation platforms as well. A similar, Python-interfaced tool is Moose (<https://moose.ncbs.res.in/readthedocs/index.html>), which can operate at multiple levels of detail ranging from biochemical reactions in subcellular components to large networks of neurons. How does BSB compare (or complement) these existing software tools? What are the new features introduced by BSB?

We thank the reviewer for the comment, a broader comparison with other modelling platforms was needed.

At a careful analysis, NeuroConstruct fulfils a similar purpose to the BSB but at the moment only supports extension in Java rather than Python or Matlab. NeuroML is a description format that does not include the algorithms to generate the data, while the BMTK/NetPyNE/BSB do. We would like to support NeuroML format in the future. Currently we allow arbitrary Arbor/NEURON/NEST single cell models in the network, while NeuroML requires existing models to be rewritten in a NeuroML description. MOOSE does not provide a full modelling framework like the BSB but focuses more on simulation backend. The most advanced feature of the BSB, to our understanding, is that we impose to the user an architecture with a code-free descriptive layer and an underlying encapsulated algorithm. Other tools either restrict access to user code or offer a library-like approach. The BSB supports both, including a mixed approach effectively combining open access to user code and APIs. Moreover, the BSB facilitates arbitrary parallelization of non-parallel user code, bringing larger scales within reach without extra code engineering efforts.

To explain this, we have extended and rephrased section 5.1.

Reviewer 1 comment 2 - Millisecond Golgi synchrony (van Welie et al)

The authors simulated two scenarios of the cerebellar subnetwork constructed by BSB. The first is the resting state activity and the second is the impulse response of the cerebellar network. The results of their model simulations are in good agreement with previously published in vitro or in vivo physiological recordings. I'm curious whether their model could also reproduce finer scale experimental findings, such as the one by van Welie and colleagues (2016, Neuron), where they show millisecond synchrony between Golgi cells in vivo, which requires gap junctions and is enhanced by sensory stimuli

We thank the reviewer for the interesting suggestion. We have used the model to analyze existing data and investigate the millisecond synchrony among GoCs. In the revised manuscript, we have included specific paragraphs in Methods (5.2.5.1), Results (3.3.2) and Discussion (4.1.2), and the new Figure 4 and Figure S3.

The initial cerebellar network reconstruction was loosely constrained by experimental measurements providing estimates of gap-junction density on GoCs. In this revision, we investigated the sensitivity of Golgi cell synchrony to gap-junction density and estimated the value required to match the experimental results reported by van Welie et al. (2016). Simulations showed that the number of gap junctions per pair had to be increased by 2.5 times compared to the prior value used in the paper. Further analysis of these results showed that the spatial organization of granular layer activity depends also on GoC coupling, which in turns depends on the distance between GoCs and on their specific morphology and orientation, supporting a modular circuit organization: a marked correlation and synchronicity can be observed within an assembly, while it tends to decrease between assemblies, indicating that GoCs coordinate segregation and integration of activities in the granular layer of cerebellum [Bisio et al 2014].

Reviewer 1 minor comments

It would be helpful to have a comprehensive supplementary table of the passive electrical properties and the active conductances of the cell types in the network model to see how these parameters compare to previously published data to achieve the reported physiological resting state activity of the cerebellar network.

We did not modify any parameters in single cell models with respect to the original papers (in the BSB, the source code of single cell models was used as such). For clarity, we have added (in the Supplementary materials) a summary table with a list of the main single cell parameters. Further properties of ionic channel models (e.g., maximum ionic conductance, reversal potential, mathematical representation) or of synaptic receptors and release properties can be found in the files in the repository (we have added the list of github links for each neuron model in the caption).

There is a typo on page 4, line 124 after the word "volume" in the references.

Thank you, typos have been fixed throughout.

Reviewer 2

The authors have developed a methodological framework utilizing the Brain Scaffold Builder to construct a 'realistic' microcircuit model of the cerebellum with all known cell components, morphologies, synaptic properties, connectivities and physiological responses. Their model contains a little over 30,000 neurons, a very small subset of the cerebellar circuit. They claim that the scope of their study is unveil the structure-function-dynamics of the cerebellar microcircuit. I am afraid they have not succeeded in their goal:

We thank the reviewer for the very useful comments that allowed us to substantially improve the manuscript. The new and the significantly revised parts are highlighted in blue.

Reviewer 2 comment 1 - Techniques used to unify experimental data

To accurately simulate the structure of a cerebellar microcircuit they utilized experimental data from various sources. The authors emphasized that by doing so the model's realism is increased. Such experimentally recorded data are full of errors, which if are not appropriately handled, but instead are used as such to build a computational model, then they are inserted into the model too. So, what techniques have the authors used to wrangle with the experimental structure data?

As it is the case of models in other brain regions [Gal et al Nat Neurosci 2017], our cerebellar model integrates sparse (anatomical and physiological) experimental data and exploits the interdependencies between microcircuit parameters to digitally reconstruct a biologically-constrained neural network. Anatomical data, such as cell types and their densities in specific layers, dendritic and axonal geometries, and synaptic densities, provided the basis for network reconstruction. Unavoidably, some parameter data were missing (never measured experimentally) or error-prone and their estimates emerged through the network reconstruction. These emergent structural parameters were then compared with available knowledge for constructive validity. For instance, the densities or ratios between the numerosity of cell populations are consistent with the number of synapses obtained from connection algorithms based on morphological intersections. Similarly, convergence, divergence, relative cell densities and positioning turned out to be properly correlated and consistent with the most advanced mouse brain atlases (Allen Brain atlas and Blue Brain atlas). We have also outlined, in the Discussion, that this reconstruction workflow is ready for update by new data as soon as they will become available.

Reviewer 2 comment 2 - functional predictions: cognition and motor control

I failed to see any functional predictions the model made. We know that the cerebellum plays functional roles in cognition and motor control. Can the simulated cerebellar microcircuit simulate any of these functionalities? If so, how does the microcircuit model computes information that support any of these functionalities? We also know that the cerebellum supports learning (LTD) via the parallel fibers and Purkinjee cell dendrites (one pathway). There are many other types of learning observed in the cerebellum. How do these forms of learning provide insightful discoveries to the cerebellum's supported functions?

We thank the reviewer for these observations. We have exploited the model to shed light on the neural correlates of behavior, by modifying specific parameters and running new simulations.

Purkinje cells provide the only output of the cerebellar cortex, and neural correlates of behaviour can be reproduced and investigated in Purkinje cells in different functional or dysfunctional conditions. We have focused on two conditions modifying PC firing patterns: (1) the knock-out of MLI inhibition on Purkinje cells, which impacts on learning in vestibulo-ocular reflex; (2) the LTD/LTP state at synapses between parallel fibers and PCs, which impacts on learning in associative tasks.

(1) The PC simple spikes reflect the integration of intrinsic pacemaker activity with excitatory and inhibitory synaptic inputs from parallel fibers and molecular layer interneurons. In the previous version we showed (in previous Fig 6c,d, actual Fig 7) that molecular layer interneurons (SCs and BCs) effectively reduced activation of PCs placed either along or beside the active *pfs*, generating feedforward and lateral inhibition. The essential role of molecular layer interneurons as regulators of cerebellar signal coding and memory formation has been proved by Wulff et al [2009]. They used a mouse line, *PC-Δγ2 KO*, in which GABA_A receptor-mediated synaptic inhibition was selectively removed from Purkinje cells, and examined how feedforward inhibition from molecular layer interneurons regulates adaptation of the vestibulo-ocular reflex. Therefore, we used our model to recreate this knock-out (KO) condition *in silico* and simulated network activity with a single impulse delivered to a bundle of 13 mfs. We have analysed the temporal dispersion (jitter) of evoked Purkinje cell simple spikes (as in the Fig. 3 of [Wulff et al 2009]). The burst response of PCs was broader and with a higher temporal dispersion (jitter) of simple spikes in KO than control condition (control: 0.49 ms; KO: 1.01 ms; $p < 0.01$ t-test) (new Fig. 8). Wulff and colleagues showed correlations between these PC activity alterations, and learning and consolidation deficits in vestibulo-ocular reflex.

(2) Furthermore, we have considered the neural correlates of Pavlovian/associative conditioning. Purkinje cell simple spike suppression is a central driving mechanism in cerebellar conditioning. Specifically, during an eye blink classical conditioning paradigm, [ten Brinke et al 2015] showed that the PCs undergo about 15% simple spike suppression from the first repetition to the last, along the learning process. This corresponds to generation of anticipated blink responses. Therefore, we have run multiple simulations with different values of synaptic conductance (g_{syn}) at AMPA-mediated pf-PC synapses. The default value of 1200 pS was lowered progressively to 30% in 10 steps. For each case, a simulation with an input of 50 Hz (lasting 160 ms) on a bundle of 4 mfs was carried out, and the firing activity modulation of each PC was extracted. Then, the amount of spike suppression was estimated as ratio between firing modulation in post-learning and the one in pre-learning. The results (new Fig. 9) show the effect on PC patterns, and therefore on number and strength of the associative responses as shown in [ten Brinke et al 2015] study. Our procedure effectively emulated the state of these plastic synapses at the end of learning, after a process of LTD at pf-PC. This model exploration allowed us to predict the amount of plasticity in the molecular layer able to accommodate the desired firing reduction of PCs, which, in turn, reflects in a proper motor learning.

These new results are now shown in Results (3.3.4, 3.3.4.1, 3.3.4.2) and explained in Methods (5.2.5.4 and 5.2.5.5). Furthermore, we added a section "4.1.3 Model predictions of neural correlates of behavior" in Discussion.

Reviewer 2 comment 3 - side-by-side comparison with experimental data

To accurately simulate the neuronal dynamics of the various cellular components of the microcircuit I would have preferred to see side-by-side comparisons of the model's outputs with experimental data under preferably the same experimental conditions. What optimization algorithm is used to fit the model's cellular responses to the experimentally recorded ones? Otherwise, if the group has not access to recorded traces of cellular responses, then perhaps they can calculate features (spike amplitude, duration, ISI, membrane tau, input resistance, etc) of the simulated cellular responses and compare them to same feature values of experimentally derived responses. This way further biological realism may be accomplished.

We have built the circuit using multi-compartment morphology-based neuron models, which were carefully validated and published previously. Each neuron and synapse model had been built by defining specific passive properties and by tuning other biophysical parameters through genetic algorithms with multi-factor cost functions targeting action potential properties, firing frequencies with different stimulation currents etc. The target values were taken from electrophysiological recordings from our and other laboratories. We did not modify any parameters of these single cell models with respect to the original papers. We have added a comprehensive table of the parameters in Supplementary materials, and the list of github links for each neuron model. We have also added a summary explanation of the construction and validation methods.

Moving on from single neuron to microcircuit scale, there are currently no experimental datasets with single-neuron resolution that include cell responses to sensory stimuli over the entire neuronal populations. Therefore, we have compared model results with the sparse experimental data that are currently available, identifying specific aspects that proved useful as validation targets (e.g. *GrCs receiving maximum excitation generated one action potential for each spike of the input burst, with short latency (< 2 ms), and faithfully followed the input up to 250 Hz*). In aggregate, this validation procedure illustrates the added value of the model, which anticipates a coherent picture of spatio-temporal cerebellar signal coding, while experimental data that are still sparse and disaggregated at the microcircuit scale. We have highlighted in the Discussion that multi-layer mesoscale data would be useful to further validate model predictions about the mechanisms of microcircuit computation in the cerebellum, e.g., following whisker stimulation. In this revision, we have also considered two conditions altering PC firing patterns (see comment n.2): (1) the knock-out of MLI inhibition on Purkinje cells, which impacts on learning in vestibulo-ocular reflex; (2) the LTD/LTP state at synapses between parallel fibers and PCs, which impacts on learning in associative tasks.

Reviewer 2 comment 4 - what previous model failures does this model overcome?

It is unclear where previous modelling attempts of the cerebellum failed, and how this model succeeded in overcoming these failures. Perhaps you may include such a section in your Discussion section

We thank the reviewer for pointing out this aspect, which was not sufficiently considered in the text. As far as we know, we have generated the most complete reconstruction of the entire cerebellar cortical microcircuit using detailed multicompartment neuron models and synapses, from the mossy fiber input to the Purkinje cell output. First among others, our model is able to mechanistically simulate signal propagation within and across the cerebellar cortical layers. In this way, it allows to monitor and control subcellular signal processing in all network

components with a resolution that goes down to ionic channel gating, dendritic processing and neurotransmitter release dynamics. For comparison we consider the main computational models of the cerebellum that have been published over the last 3 decades, and we have added “4.1.4 Comparison with previous cerebellar models” in Discussion.

The current model of the cerebellar cortical network integrates and extended all the previous realizations by featuring an integrated reconstruction and simulation strategy, using multi-compartment neurons with cell-specific membrane mechanisms, using synapses with intrinsic neurotransmitter release dynamics and short-term plasticity, and adopting multiple connection rules including morphology-based touch-detection and voxel-intersection. These advancements reflect into the ability of the model to capture a large set of biological properties of the network under various physio-pathological conditions (e.g. new Fig 4 about the GoC synchrony, new Fig.s 8 and 9 about neural correlates of behavior).

REVIEWERS' COMMENTS:

Reviewer #1 (Remarks to the Author):

This revised version of the MS has addressed all the concerns I had raised with additional simulations, analyses and re-writing. It is therefore significantly improved and I have no additional comments.